# NextCoder: Robust Adaptation of Code LMs to Diverse Code Edits

Tushar Aggarwal [* 1]   Swayam Singh [* 1]   Abhijeet Awasthi [1]   Aditya Kanade [1]   Nagarajan Natarajan [1]

## Abstract

Software engineering activities frequently involve edits to existing code. However, contemporary code language models (LMs) lack the ability to handle diverse types of code-edit requirements. In this work, we attempt to overcome this shortcoming through (1) a novel synthetic data generation pipeline and (2) a robust model adaptation algorithm. Starting with seed code examples and diverse editing criteria, our pipeline generates high-quality samples comprising original and modified code, along with natural language instructions in different styles and verbosity. Today's code LMs come bundled with strong abilities, such as code generation and instruction following, which should not be lost due to fine-tuning. To ensure this, we propose a novel adaptation algorithm, SeleKT, that (a) leverages a dense gradient-based step to identify the weights that are most important for code editing, and (b) does a sparse projection onto the base model to avoid overfitting. Using our approach, we obtain a new series of models NextCoder (adapted from QwenCoder-2.5) that achieves strong results on five code-editing benchmarks, outperforming comparable size models and even several larger ones. We show the generality of our approach on two model families (DeepSeekCoder and Qwen-Coder), compare against other fine-tuning approaches, and demonstrate robustness by showing retention of code generation and general problem-solving abilities post adaptation. We opensource the models, synthetic dataset, and implementation at aka.ms/nextcoder.

---

[*]Equal contribution  [1]Microsoft Research India. Correspondence to: Tushar Aggarwal <tushar.aggarwal53@gmail.com>, Swayam Singh <t-swsingh@microsoft.com>, Abhijeet Awasthi <abawasthi@microsoft.com>, Aditya Kanade <kanadeaditya@microsoft.com>, Nagarajan Natarajan <nagarajn@microsoft.com>.

*Proceedings of the 42^nd International Conference on Machine Learning*, Vancouver, Canada. PMLR 267, 2025. Copyright 2025 by the author(s).

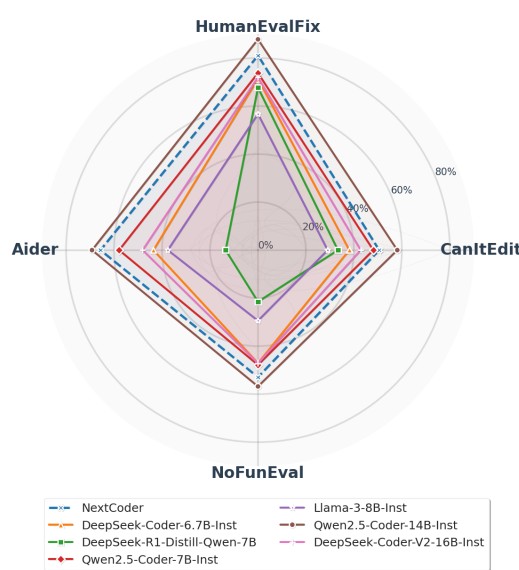

*Figure 1.* Performance of state-of-the-art code LMs, in the parameter range 6.7B-16B, on code editing benchmarks. NextCoder-7B is our code-editing model with QwenCoder-2.5-7B as the base, fine-tuned using the proposed SeleKT algorithm on synthetic and real code editing tasks. For NoFunEval, we consider instances with binary oracles to ensure consistency with other benchmarks. We present detailed results in Section 5.2, Table 4.

## 1. Introduction

Code editing is a fundamental ability with pervasive use in automating software engineering activities. Recent benchmarks reveal that contemporary code language models (LMs), particularly the smaller and open-weight LMs, struggle to edit code based on natural language instructions (Muennighoff et al., 2023; Guo et al., 2024b; Cassano et al., 2023; Singhal et al., 2024). This is despite many of them incorporating commit-data from GitHub in pre-training (Li et al., 2023; Lozhkov et al., 2024) or fine-tuning (Muennighoff et al., 2023; Cassano et al., 2023; Xie et al., 2025), where the commit messages are used as instructions.

In this work, we aim to enhance the ability of code LMs to handle diverse types of code-edit requirements. This poses two challenges: (1) lack of high-quality fine-tuning data and (2) the risk of losing the strong and general abilities (such as code comprehension, code generation and instruction following, acquired during pre-training and instruction tuning)

---

due to catastrophic forgetting (Goodfellow et al., 2013).

We address both these challenges in this paper. We propose (1) *a synthetic data generation pipeline* which starts with seed code examples and diverse code editing aspects to generate samples comprising original and modified code, along with natural language instructions. The code editing aspects are at a high-level and simply state dimensions along which samples should be generated (e.g., bug fixing, runtime improvement, etc.). Our multi-stage pipeline generates original code which is intentionally deficient in some dimension so that a meaningful edit that addresses the deficiency can be generated. Inspired by (Wei et al., 2024b), seed code is used to ensure diversity in the generated code and its scale (ranging from functions, classes, to files). It is important to support diversity in prompting styles as well. Our instruction generation stage samples natural language edit instructions, that are well-fitted to the change from the original to the modified code, in different styles (instruction versus conversation) and verbosity (concise versus detailed). We generate synthetic data using GPT-4o and Llama-3.3-70B models, and use it together with high-quality commit data from CommitPackFT (Muennighoff et al., 2023).

We observe that supervised fine-tuning on the curated data hampers pre-learned abilities of code LMs like code generation. We therefore (2) devise *a robust adaptation algorithm*, called "selective knowledge transfer" (SeleKT), which selectively adjusts the base model weights with respect to a subset of fine-tuned model weights. Unlike existing model adaptation techniques (Nguyen et al., 2024a) which select the updatable weights *a priori*, we select the weights periodically based on their magnitude of change during fine-tuning. (3) We experimentally show superiority of our model adaptation algorithm over existing methods.

Putting all of these together, (4) we construct a new series of models NextCoder, adapted from QwenCoder-2.5 instruct variants fine-tuned on eight programming languages, that achieves strong results on five code-editing benchmarks. These benchmarks cover multiple programming languages and test a variety of scenarios, including function, class or file level edits, code improvements, and bug fixing. As shown in Figure 1, NextCoder-7B consistently outperforms models of comparable size. On several tasks, it even outperforms larger models such as DeepSeekCoder-V2-16B, DeepSeekCoder-33B and Llama-3-70B. We further show that (5) our approach generalizes to other model families by improving the code editing performance of DeepSeekCoder-6.7B. To illustrate the robustness of our SeleKT approach, (6) we show that unlike full fine-tuning (SFT), LoRA (Hu et al., 2021) or a model merging approach TIES (Yadav et al., 2024), the performance of models fine-tuned using our method does not degrade their code generation abilities. (7) We demonstrate the effectiveness of our approach on dif-

ferent model sizes by finetuning 3B, 14B, and 32B variants of the QwenCoder-2.5 instruct model. On the challenging Aider Polyglot (Gauthier, 2024b) benchmark, NextCoder-32B outperforms several strong models such as GPT-4o (2024-11-20), DeepSeek-V2.5 and QwenCoder-2.5-32B.

In summary, we make the following contributions:
**1. Synthetic pipeline for diverse code-editing examples:** We present a multi-stage pipeline and sample 127K high-quality, diverse code editing examples (comprising 229M tokens), where the diversity comes from multiple dimensions: (a) granularity of code, (b) types of code-editing requirements, (c) the style and verbosity of natural language instructions, and (d) choice of programming language.
**2. Robust adaptation algorithm:** To prevent catastrophic forgetting due to fine-tuning, we propose an algorithm, SeleKT, which only selectively updates model weights. We demonstrate that this helps retain the code generation abilities after fine-tuning on code-editing data.
**3. Strong code-editing models:** We demonstrate significant improvements in code-editing performance for models like QwenCoder-2.5 and DeepSeekCoder, showing that SeleKT outperforms full and parameter-efficient finetuning methods across four code-editing benchmarks. Our NextCoder-7B derived from Qwen2.5-Coder-7B, outperforms other models of comparable size and even matches larger models across multiple tasks (Figure 1). On the popular Aider and Aider Polyglot benchmarks, NextCoder-32B is SOTA against open-source models up to 236B parameters.
**4. Opensource:** We opensource the models, synthetic dataset, and implementation at aka.ms/nextcoder.

## 2. Related Work

**Language Models for Code Editing** Language models of code demonstrate varying levels of proficiency in following code-editing instructions, as measured by benchmarks such as CanItEdit (Cassano et al., 2023), NoFunEval (Singhal et al., 2024), SWE-Bench (Jimenez et al., 2023), Aider (Gauthier, 2024a), CodeEditorBench (Guo et al., 2024b), EditEval (Hu et al., 2023), and RES-Q (LaBash et al., 2024). Prior research has explored ways to specialize language models for code editing. For example, StarCoder (Li et al., 2023), OctoCoder (Muennighoff et al., 2023), and EditCoder (Cassano et al., 2023) leverage git commits as part of their pre-training (Li et al., 2023) or fine-tuning datasets (Muennighoff et al., 2023; Cassano et al., 2023) to enable models to edit source files based on natural language commit messages. Similarly, StarCoder2 (Lozhkov et al., 2024) incorporates GitHub issues, pull requests, and associated files containing code edits, potentially equipping the model with code-editing capabilities during pre-training. SWE-Fixer (Xie et al., 2025) specializes the Qwen2.5-72B model for code editing by fine-tuning it on GitHub issues. We

focus on generating synthetic data that samples diverse code edits and on robust adaptation of pre-trained models through fine-tuning on such data.

**Synthetic Generation of Coding Data** Generating synthetic instruction-response pairs has become a standard approach for post-training alignment of models to follow instructions (Wei et al., 2024a; Wang et al., 2023; Xu et al., 2023; Luo et al., 2023; Zhao et al., 2024; Li et al., 2024). These methods have been extended to code LMs. For example, WizardCoder (Luo et al., 2023) employs the EvolInstruct framework (Xu et al., 2023) to enhance the complexity of instructions in the CodeAlpaca dataset (Chaudhary, 2023), creating a more challenging instruction-following dataset. The CodeAlpaca dataset itself was synthetically generated using the Alpaca (Taori et al., 2023) and Self-Instruct (Wang et al., 2023) pipelines. Both EvolInstruct and CodeAlpaca primarily focus on function-level coding tasks, with limited coverage of code-editing problems.

A more recent class of methods condition the example generation process on seed-code derived from real code files. Methods such as Self-CodeAlign (Wei et al., 2024a), Wave-Coder (Yu et al., 2024), and OSS-Instruct (Wei et al., 2024b) belong to this category. These approaches significantly improve task and instruction diversity by leveraging diverse seed codes. However, they remain untested for generating file-level coding examples that include multiple classes or functions, and they do not emphasize code-editing tasks. To the best of our knowledge, InstructCoder (Hu et al., 2023) is the only method explicitly designed to generate diverse synthetic data for code-editing tasks by conditioning on seed examples. Nevertheless, the examples produced by the InstructCoder pipeline are limited to short, function-level code snippets and are restricted to Python. In contrast, our method supports the generation of both function/class-level and file-level code-editing examples across multiple task categories and programming languages.

**Overcoming Catastrophic Forgetting and Robust Fine-tuning** We find that fine-tuning LMs on code-editing examples worsen their performance on code-generation tasks, in line with the catastrophic forgetting phenomenon (Goodfellow et al., 2013; Kirkpatrick et al., 2017). Model-merging (Xiao et al., 2023; Morrison et al., 2024; Yadav et al., 2024) has recently emerged as a method for learning new tasks while avoiding catastrophic forgetting of knowledge acquired during pre-training. The sparse adaptation technique of Nguyen et al. (2024b) selects the parameters to be fine-tuned based on the top-k components of the task vector, similar to our method. However, they do this *a priori*, and then only fine-tune the selected parameters (i.e., sparse gradients only). In contrast, our algorithm periodically reassesses the parameters, and performs full fine-tuning of the entire model (**Dense Gradients** step in Algorithm 1).

| Language | GPT-4o | Llama-3.3-70B | Total | Tokens(M) |
|---|---|---|---|---|
| Python | 8406 | 6963 | 15279 | 28.63 |
| C | 7039 | 10114 | 17153 | 33.48 |
| C++ | 6272 | 11065 | 17337 | 30.93 |
| Java | 6447 | 9881 | 16328 | 27.61 |
| JS | 7367 | 8663 | 16030 | 25.92 |
| Rust | 4701 | 11737 | 16438 | 30.43 |
| Go | 4503 | 10701 | 15204 | 28.56 |
| Kotlin | 3470 | 9802 | 13272 | 22.16 |
| Total | 48205 | 78926 | 127041 | 227.72 |

*Table 1.* Number of synthetic examples and tokens generated per programming language and model type.

## 3. Synthesizing a Diverse Code Editing Dataset

Git commits promise to be a readily available source of supervision for adapting models for file-level code editing, as also explored by OctoCoder (Muennighoff et al., 2023). However, in our preliminary experiments (Table 8), we find that fine-tuning on source-target file pairs from Commit-PackFT (Muennighoff et al., 2023), a dataset derived from GitHub commits, with commit messages as instructions, yields limited improvements in code editing. We attribute this to the generally limited quality and diversity of data on GitHub and the lack of informative commit messages.

Thus, in addition to using CommitPackFT as a source of supervision, we propose a novel method for generating high-quality code-editing examples starting with real seed data from GitHub using large and medium LMs, such as GPT-4o and the instruct version of Llama-3.3-70B. Our method provides greater diversity by offering explicit control over varying levels of granularity (function, class, and file-level code edits), a wide range of code edit types (e.g., bug fixing, latency and runtime improvements, addressing security vulnerabilities, optimizing resource utilization, and enhancing maintainability), diversity in programming languages, and varying levels of instruction complexity (both concise and verbose, and single-turn as well as multi-turn style conversations). Table 1 provides statistics of the code editing data generated using our method across eight programming languages. We generate approximately 127K examples and 229M tokens, with 48K examples produced using GPT-4o. We perform contamination analysis of our generated data with respect to the evaluation benchmarks in Section A.1.5.

Figure 2 illustrates our data generation pipeline and Figure 9 shows an example seed code passed through GPT-4o in our pipeline, which comprises four main components: (i) Problem and Source Code Generation, (ii) Target Code Generation, (iii) Instruction Generation, and (iv) Quality-Based Filtering. Each component is described below.

**i) Problem and Source Code Generation** The LLM used for synthetic-data generation is prompted to generate the problem description and source code as solution,

*Figure 2.* **Our synthetic data generation pipeline**: The input to the pipeline is a **seed code** snippet, **modularities** (function, class or file) which defines the scope of the output code and **aspects** to improve up on (latency, resource utilization, runtime efficiency, maintainability, security, and general improvements along with bug fixing). The output is a synthetic example, approved by the final quality checker, consisting of **problem statement**, **source code**, **target code**, and **instructions** in different styles and verbosity (detailed, concise, human-like, conversational). The details of the pipeline stages are presented in the running text.

which is conditioned on required code modularity (function-level/class-level/file-level) and the provided seed code sampled from the StarCoder dataset (Kocetkov et al., 2022; Li et al., 2023), where we sample only from files that have more than 10 lines and contains logic like loop, functions, conditional statements or classes. The generated source code contains flaws that align with the identified improvement areas (e.g., Bug Fixing, Improving Latency, Optimizing Resource Utilization) in the prompt. Additionally, the LLM generates metadata that outlines the specific flaws present in the code. Each instance of the synthetic data is generated using a single seed code. The prompt used for this step is given in Figure 5.

**ii) Target-Code Generation** Next, we prompt the LLM to generate the target code conditioned on the problem description, source code and the metadata produced in the previous step. We design this prompt to also output an explanation of the edits made by the model to source code to obtain target code. The prompt used for this step is given in Figure 6.

**iii) Instruction Generation** In the subsequent step, we prompt the LLM to generate code-editing instructions using the source code, target code, and the editing explanation generated in previous steps as input. A parameter in the prompt specifies whether the instruction should be in Concise, Detailed, Human, or Conversational format. Concise instructions are high-level, often under-specified, three-line descriptions that do not explicitly detail the required changes. Detailed instructions are more verbose and provide specific information about the required changes, such as specifying the exact function to be modified for improved runtime. Human instructions are very brief, informal, and natural-language-based messages that are typically 1-2 sentences long and provide a high-level overview of the necessary changes without going into technical detail (e.g., "Hey modify the given code to improve its runtime"). Conversational instructions represent a user-assistant interaction in a chat format, where the user sequentially specifies the required changes. Each instance therefore offers four fine-tuning examples, one for each format. The prompt used for this

step is given in Figure 7.

**iv) Quality-based Filtering** Finally, to ensure high quality of generated examples, we prompt the LLM to verify whether the target file is a correctly edited version of the source file, consistent with the instructions generated in the previous step. The model evaluates the instance by assigning a score from 0 to 10 across five criteria: (1) Correctness of edits w.r.t. requested improvements (e.g. latency), (2) adherence of the edits to the instructions, (3) code quality, (4) instruction quality, and (5) the usefulness of the example for fine-tuning small models. An instance is deemed valid if its average score is at least 7, and all individual scores exceed 5. The prompt used for this step is given in Figure 8. We also performed a human study to assess the quality of the generated data, and the details are in Section A.3.

## 4. Robust Model Adaptation

Pre-training LMs on large amounts of data followed by fine-tuning them on relatively smaller amounts of data from downstream scenarios is now a standard practice. But naïve fine-tuning can result in poor generalization performance.

The state-of-the-art code LMs such as DeepSeekCoder and Qwen already have gone through a rigorous pipeline of pre-training and instruction-tuning, on vast amounts of code and text tokens (as many as 5.5T tokens (Hui et al., 2024), in the case of Qwen-2.5). At the same time, these pre-trained LMs find several code-editing scenarios challenging (as we show in Section 5.2). So, the key question is how we can strike a good balance between task-specific (i.e., code editing scenarios) performance and the generalization abilities of the pre-trained model (i.e., code comprehension, instruction-following, code generation, etc.). In this section, we present our technique for *robustly adapting* code LMs to diverse code editing tasks.

**Robust Adaptation Problem** Formally, we want to fine-tune a given code LM, denoted by $\theta_{\text{base}}$, such that: (a) the resulting LM $\theta_{\text{FT}}$, has improved code editing abilities, as

determined by the training loss on the data presented in the previous section, (b) while preserving the generalization abilities of the base LM (as determined by the performance on real-world code benchmarks). The robust adaptation problem (Wortsman et al., 2022; Tian et al., 2023) can be posed as:

$$\arg\min_\theta \mathcal{L}(\theta) \quad \text{s.t.} \quad \|\theta - \theta_{\text{base}}\| \le c\,, \qquad (1)$$

where $\mathcal{L}$ denotes the next-token prediction loss on the training data, $\|\cdot\|$ is a suitable norm, and $c$ is a constant.

**Inadequacy of Existing Solutions** State-of-the-art techniques for adaptation (robust or otherwise) largely follow the parameter-efficient fine-tuning (PEFT) paradigm. That is, fine-tuning is localized to a small fraction of the parameters of the base model. For instance, the widely-used LoRA technique (Hu et al., 2021) fine-tunes a small number of parameters added to the base LM, keeping the entirety of the base LM frozen otherwise. More recent techniques carefully select the parameters to fine-tune, e.g., a few layers (Lee et al., 2023) or a few parameters across the layers (Nguyen et al., 2024b). The crux of these techniques seems to be that: (**A1**) focus on fine-tuning a small number of parameters to avoid overfitting on a small amount of training data, and (**A2**) fix the parameters to be fine-tuned *a priori*, before the training even *begins*. Although the design choice (**A1**) is reasonable and is arguably prudent, we question the design choice (**A2**) of these techniques.

While recent robust adaptation techniques try to achieve the model accuracy-efficiency trade-off using sparse gradients in their updates, the appeal for efficiency during fine-tuning seems to be *at cross* with the desire to also achieve low generalization error and out-of-domain robustness. We show this empirically in our experiments (Table 4); Hu et al. (2021) also observe that there is some degradation in performance due to the choices made in LoRA.

Our **key insights** are simple:
(**1**) What parameters need to be fine-tuned for the task(s) at hand **should be continuously re-assessed** conditioned on the difficulty of the downstream scenario as dictated by the fine-tuning data and the training loss.
(**2**) (**Dense Gradients**) We can update all the model parameters to determine the direction of parameter changes that best minimizes the training loss on the code editing tasks, unlike PEFT methods, and then (**Sparse Projection**) compute a suitable projection to ensure that the parameters are guaranteed to be close to the base model (Figure 3).

**Proposed Solution** To implement the two insights above, we choose the $L_0$-norm in (1), i.e., we want the updates to be localized to a small set of parameters of the base model. The $L_0$-norm makes the projection step computationally easy: we first compute dense gradients by doing full fine-

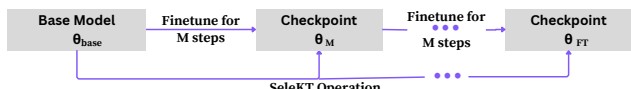

*Figure 3.* Proposed adaptive fine-tuning technique SeleKT.

---

**Algorithm 1** SeleKT: **Sele**ctive **K**nowledge **T**ransfer

---

**Require:** Base LM weights $\theta_{\text{base}}$, training data $\mathcal{D}$, epochs $E$, periodicity $M$, sparsity $\alpha$.
**Ensure:** Final fine-tuned weights $\theta_{\text{FT}}$.
1: Initialize $\theta \leftarrow \theta_{\text{base}}$.
2: **for** epoch $e = 1$ to $E$ **do**
3:     **for** each minibatch $\mathcal{D}[s]$ **do**
4:         $\theta \leftarrow \text{TrainStep}(\theta, \mathcal{D}[s])$    **[Dense Gradients]**
5:         **if** $s \mod M = 0$ **then**
6:             Compute task vector: $\tau \leftarrow \theta - \theta_{\text{base}}$
7:             Select top-$\alpha N$ parameters:

$$\gamma[i] = \begin{cases} 1, & i \in \text{top-k}(|\tau|, \lfloor \alpha \cdot N \rfloor) \\ 0, & \text{otherwise} \end{cases}$$

8:             $\theta \leftarrow \theta_{\text{base}} + \gamma \odot \tau$    **[Sparse Projection]**
9:         **end if**
10:     **end for**
11: **end for**
12: **return** $\theta$ as $\theta_{\text{FT}}$.

---

tuning of the model $\theta$, and the compute the top-k non-zero entries (by magnitude) on the (accumulated) gradient vector or the "task vector" $\theta - \theta_{\text{base}}$. This also ensures that the parameter selection is global and not confined to specific layers or other heuristics employed in earlier robust fine-tuning strategies (Lee et al., 2023). The resulting "selective knowledge transfer" problem is:

$$\arg\min_\theta \mathcal{L}(\theta) \quad \text{s.t.} \quad \|\theta - \theta_{\text{base}}\|_0 \le c\,. \qquad (2)$$

Our algorithm, SeleKT, short for **Sele**ctive **K**nowledge **T**ransfer, is presented in Algorithm 1. It is parameterized by (i) sparsity $\alpha$, or the fraction of the total number of model parameters $N$ to be updated, and (ii) periodicity $M$, or how often the projection step needs to be performed.

**Choice of Training Loss $\mathcal{L}$ in** (2) We use cross-entropy loss for next token prediction as the objective function in our experiments. For examples formatted as (instruction, response) pairs, we apply the loss on the entire example. For examples formatted as multi-turn conversations (Section 3), we apply the loss only to the final response generated by the model and not on the conversations.

**Lemma 1.** *For any given a base LM $\theta_{base}$, and for the*

| Method | HumanEvalFix | CanItEdit | Aider |
|---|---|---|---|
| Update $\theta_{\text{base}}$ | 79.5 | 48.0 | 55.6 |
| Fix $\theta_{\text{base}}$ | **81.1** | **50.5** | **65.7** |

*Table 2.* Performance of SeleKT with and without periodically updating the base model QwenCoder-2.5-7B as in Remark 2.

| Benchmark | Description | Examples |
|---|---|---|
| CanItEdit | Bug fixing (class-level) | 210 |
| HumanEvalFix | Bug fixing (function-level) | 164 |
| NoFunEval | Code improvements (file-level) | 397 |
| Aider | Bug fixing (conversational, file-level) | 133 |
| Aider Polyglot | Bug fixing (conversational, file-level) | 225 |
| HumanEval+ | Code generation (function-level) | 164 |
| MBPP+ | Code generation (function-level) | 378 |
| GSM8K | Mathematical problem solving | 1.32K |
| MMLU | Multiple-choice STEM knowledge | 3.15K |

*Table 3.* **Evaluation datasets**: We use four diverse code editing benchmarks with varying input granularity and code-editing criteria. Additionally, we also evaluate on two code generation tasks and two non-coding tasks.

*setting $\alpha = c/N$, where $N$ is the model size, the fine-tuned model $\theta_{FT}$ satisfies the constraint in the objective* (2).

The proof is straight-forward as (i) in Step 6 of Algorithm 1, we always compute the task vector with respect to the base model, (ii) the mask vector computed in Step 7 selects $c$ coordinates. Together with the update in Step 8, the constraint is guaranteed to satisfy.

**Remark 1** (Efficiency)**.** While the projection step itself is straight-forward, computation of dense gradients can be more expensive. The cost is mitigated to some extent by doing mini-batching and restricting the total number of epochs to a few. We show in Sections 5.2 and 5.4 that the resulting improvements in generalization accuracy are significant.

**Remark 2** (Alternative Update)**.** An alternative style of update in Algorithm 1 is to periodically also update the base model, i.e. $\theta_{\text{base}} \leftarrow \theta$ after Step 8. So the future task vector (in Step 6) will be computed with respect to the updated base model. This update style also guarantees that the final $\theta_{FT}$ will be close to the base model in the $L_0$-norm sense. To be precise, we can show, an albeit weak bound, $\|\theta_{FT} - \theta_{\text{base}}\|_0 \leq c \cdot E \cdot |\{\mathcal{D}[s]\}|/M$, where $|\{\mathcal{D}[s]\}|$ is the number of mini-batches, $M$ is periodicity, for a sufficiently small choice of $\alpha$. We also find that this style (denoted by "Update $\theta_{\text{base}}$") performs worse empirically, from Table 2 (details of fine-tuning and benchmarks in Section 5.1).

# 5. Experiments

We present the details of our experimental setup, followed by a discussion of the main results and an ablation study.

## 5.1. Experimental Setup

**Fine-tuning Dataset** In addition to the synthetic data (Table 1), we used 127K instances from CommitPackFT to fine-tune our models. CommitPackFT, restricted to the eight languages (Table 1), consists of 153K real GitHub commits. We filtered out the examples from the dataset which do not have source code (e.g., edits only to config files) to retain only code-editing instances (Table A.1).

**Our Models and Baselines** For fine-tuning, we consider the instruct versions of DeepSeekCoder-6.7B (Guo et al., 2024a) and Qwen2.5-Coder-7B (Hui et al., 2024). To demonstrate effectiveness of our approach across model sizes, we also fine-tuned the 3B, 14B, and 32B variants of the

QwenCoder-2.5 instruct model. We compare our fine-tuned models against models from DeepSeekCoder, Llama, and Qwen families in the range 6.7B to 70B parameters. We also compare with GPT-4o (Hurst et al., 2024) and the Qwen-7B distilled from the latest, reasoning-enhanced DeepSeek-R1 model (Guo et al., 2025). For Aider Polyglot, we compare against leading models from their public leaderboard.

**Hardware** For fine-tuning and inference, we use 8 NVIDIA H100 GPUs, each with 80GB of VRAM. For data generation using GPT-4o (version 2024-05-13), we use the OpenAI API. Fine-tuning takes about six hours per epoch of wall-clock time. Following Singhal et al. (2024), we perform run-time evaluations for NoFunEval on an Azure NC16 VM (NC16).

**Implementation** We fine-tune for 3 epochs, across all our experiments, using AdamW optimizer (Loshchilov & Hutter, 2017) with a learning rate of $10^{-5}$, and a WarmupLR scheduler (Kim et al., 2021) with a warmup ratio of 0.1. For efficient memory management, we use sample packing with a maximum sequence length of 8192 tokens for DeepSeekCoder-6.7B and 16384 tokens for QwenCoder variants. We initialize the models from their respective pre-trained checkpoints (instruct versions), and fine-tune them on their respective chat templates (HuggingFace, 2025). Appendix A.1.2 provides additional implementation details.

**Hyperparameters** (i) We used a temperature of 0.6 for data generation for both Llama-3.3-70B and GPT-4o. (ii) We fix the periodicity to 1 epoch in the SeleKT algorithm unless specified otherwise, i.e., $M = $ total number of minibatches. We set sparsity $\alpha = 0.05$ per layer. We selected these values based on initial experiments. In Section 5.7, we show ablations on these choices. Additional hyperparameter tuning for SFT and LoRA are detailed in Section A.1.4.

**Evaluation Datasets and Metrics** Table 3 presents the diverse datasets we use for evaluating our and baseline models. Specifically, we use CanItEdit (Cassano et al., 2023), HumanEvalFix (Muennighoff et al., 2023), NoFunEval (Singhal et al., 2024), Aider (Gauthier, 2024a) and Aider Polyglot (Gauthier, 2024b) to evaluate on code-editing tasks. We

| Models | NoFunEval | | | | | HumanEvalFix | Aider | CanItEdit |
|---|---|---|---|---|---|---|---|---|
| | Latency | Res. Util. | Runtime Eff. | Maintain. | Security | | | |
| GPT-4o | 45.6* | 39.3* | 3.389* | 57.6 | 55.1 | 90.2* | 74.4 | 59.5 |
| Llama-3-70B-Inst | 34.4 | 28.3 | 2.816 | 43.7 | 58.1 | 77.4 | 51.1 | 56.7 |
| DeepSeekCoder-33B | 30.0 | 24.0 | 2.589 | 38.0 | 53.9 | 74.4 | 58.6 | 49.5 |
| DeepSeekCoder-V2-16B | 23.6 | 21.4 | 2.274 | 37.5 | 54.7 | 72.0 | 48.1 | 42.8 |
| QwenCoder-2.5-32B | 42.1 | 36.9 | 3.006 | 64.0* | 58.6 | 90.2* | 75.2* | 60.9* |
| QwenCoder-2.5-14B | 38.1 | 32.2 | 2.597 | 50.7 | 55.8 | 87.8 | 66.9 | 58.1 |
| Llama-3-8B-Inst | 22.5 | 18.7 | 1.255 | 20.6 | 55.1 | 56.7 | 39.8 | 29.0 |
| DeepSeek-R1-Qwen-7B | 14.4 | 8.8 | 1.185 | 9.6 | 41.2 | 67.7 | 13.5 | 33.3 |
| DeepSeekCoder-6.7B | 20.5 | 21.0 | 2.275 | 35.3 | 61.8 | 71.3 | 43.6 | 38.1 |
| DeepSeekCoder-6.7B-LoRA | 21.0 | 18.0 | 1.245 | 28.3 | 54.0 | 70.7 | 41.4 | 37.2 |
| DeepSeekCoder-6.7B-SFT | 22.4 | 15.0 | 1.359 | 23.8 | 57.2 | 65.2 | 30.8 | 41.4 |
| DeepSeekCoder-6.7B-TIES | 22.1 | **25.3** | 2.166 | 37.6 | 62.9 | 73.8 | 48.1 | 45.7 |
| DeepSeekCoder-6.7B-SeleKT | 24.8 | 22.0 | 2.150 | 40.0 | 63.6 | 76.0 | 47.1 | 49.6 |
| QwenCoder-2.5-7B | 26.5 | 24.7 | 2.190 | 39.3 | 58.7 | 73.8 | 59.4 | 48.1 |
| QwenCoder-2.5-7B-LoRA | 26.9 | 21.7 | 2.133 | 38.2 | 55.0 | 70.7 | 40.6 | 44.3 |
| QwenCoder-2.5-7B-SFT | 25.1 | 22.5 | 1.387 | 30.6 | 54.8 | 70.1 | 48.9 | 36.7 |
| QwenCoder-2.5-7B-TIES | **27.8** | 24.9 | 2.180 | 44.4 | 60.1 | 79.5 | 60.2 | 47.0 |
| QwenCoder-2.5-7B-SeleKT (NextCoder) | 26.4 | 23.2 | **2.286** | **46.1** | **65.3 *** | **81.1** | **65.7** | **50.5** |

*Table 4.* **Performance of baseline and fine-tuned code LMs on code-editing benchmarks**: The numbers (the higher the better) denote score for NoFunEval, and % accuracy for all others. For NoFunEval, we considered the scores for the best (Max) performing prompt. The rows shaded blue are the models obtained using our approach; the rows shaded gray are baseline models of comparable sizes. Best number in the comparable group is in **bold**, and overall best is indicated by star *.

choose these datasets to ensure diversity in terms of (a) **scale** (class or function or file level), (b) **requirements** (functional improvements such as bug fixing and non-functional improvements such as runtime efficiency and security), (c) **instruction details** (terse vs. detailed), and (d) **scenarios** (standard vs. chat-based). Each dataset comes with its own metrics to automatically evaluate the model outputs. While the common metric is execution accuracy on test cases, the NoFunEval benchmark also utilizes runtime improvements, static analysis checks, and DiffBLEU scores (Bairi et al., 2023) for evaluation. We also evaluate the larger 14B and 32B model variants on Aider Polyglot benchmark (Gauthier, 2024a). This benchmark has gained popularity as it contains particularly challenging problems across multiple programming languages, proving difficult even for very large models. Notably, even advanced models like GPT-4o (2024-11-20) solve only 18.2% of the problems in this benchmark.

Additionally, we also evaluate the models on (a) standard code generation benchmarks, HumanEval+ and MBPP+ (Liu et al., 2023) (Table 3, rows 6–7), to measure the extent to which models fine-tuned for code editing still retain their code generation abilities. These benchmarks require a model to complete a function given its signature and a short description. The evaluation metric is execution accuracy on test cases. Further, we do evaluation on (b) non-coding benchmarks, GSM8K (Cobbe et al., 2021) and STEM subset of MMLU (Hendrycks et al., 2021) (Table 3, rows 8–9), to gauge how well they maintain math problem solving and natural-language understanding capabilities. Appendix A.1.3 provides details on the benchmarks.

## 5.2. Performance on Code-Editing Benchmarks

Table 4 presents the main results of our experiments covering four well-known benchmarks for code editing. We compare (i) our adapted models with existing language models for code (ii) SeleKT, our adaptation method, with standard model adaptation methods like supervised fine-tuning (SFT), LoRA (Hu et al., 2021) and TIES (Yadav et al., 2024). In Table 4, our SeleKT based adaptations (DeepSeekCoder-6.7B-SeleKT and QwenCoder-2.5-7B-SeleKT) are highlighted in blue . Rows in Table 4 highlighted in gray refer to various similarly sized models in the parameter range 6.7B to 8B. Our main observations can be summarized as follows.

**i) SeleKT provides consistent gains over the original instruct models.** For example, DeepSeekCoder-6.7B-SeleKT outperforms DeepSeekCoder-6.7B on all tasks except the run-time improvement task in NoFunEval. Interestingly, on CanItEdit benchmark, DeepSeekCoder-6.7B-SeleKT provides 11.5 point gains in accuracy over DeepSeekCoder-6.7B. Similarly, NextCoder-7B outperforms QwenCoder-2.5-7B on all tasks except for Latency and Resource Utilization in NoFunEval.

**ii) SeleKT frequently outperforms the standard model adaptation methods.** Surprisingly, supervised fine-tuning of models like QwenCoder-2.5-7B frequently resulted in worse performance across various benchmarks w.r.t. the original model ( QwenCoder-2.5-7B-SFT vs QwenCoder-2.5-7B in Table 4), indicating overfitting. To address this problem, we designed SeleKT (Section 4), for robust model adaptation. We see that SeleKT outperforms parameter-efficient fine-tuning methods like LoRA (Hu et al., 2021)

and model-merging methods like TIES (Yadav et al., 2024), often used for adapting models while limiting loss in prior knowledge (NextCoder-7B vs QwenCoder-2.5-7B-LoRA vs QwenCoder-2.5-7B-TIES in Table 4).

**iii) SeleKT provides best code-editing performance among models of its size.** Baseline models of comparable size include a reasoning based DeepSeek-R1-Qwen-7B (Guo et al., 2025), Llama-3-8B-Inst (Dubey et al., 2024), DeepSeekCoder-6.7B (Guo et al., 2024a) and its adaptations, QwenCoder-2.5-7B (Hui et al., 2024) and its adaptations. We observe that our best model QwenCoder-2.5-7B-SeleKT, referred as NextCoder-7B, has the best overall performance across all models in this parameter range.

**iv) Performance of adapted SeleKT models depends on the performance of their unadapted versions.** We observe that QwenCoder-2.5-7B and NextCoder-7B generally outperform DeepSeekCoder-6.7B and DeepSeekCoder-6.7B-SeleKT, respectively. This suggests that higher-performing models lead to more accurate adaptations, even when model sizes are similar.

**v) Comparison with larger models.** Notably, NextCoder-7B matches or even surpasses larger models on many tasks. For instance, it outperforms DeepSeekCoder-V2-16B (twice its size) on all tasks. Similarly, NextCoder-7B clearly outperforms Llama-3-70B-Inst and DeepSeekCoder-33B on HumanEvalFix, Aider, and the Maintainability and Security splits of NoFunEval. As expected, much stronger models like GPT-4o and QwenCoder-2.5-32B substantially outperform smaller models, including our SeleKT models.

In Appendix C, we present qualitative examples on difference between the base model and our fine-tuned version.

### 5.3. Effectiveness of SeleKT on Different Model Sizes

| Models | HumanEvalFix | CanItEdit | Aider | Aider Polyglot |
|---|---|---|---|---|
| QwenCoder-2.5-3B | 73.2 | 37.1 | 36.8 | - |
| QwenCoder-2.5-3B-LoRA | 64.6 | 36.2 | 35.8 | - |
| QwenCoder-2.5-3B-SFT | 76.2 | 32.4 | 30.1 | - |
| NextCoder-3B | 75.6 | 42.4 | 37.6 | - |
| QwenCoder-2.5-14B | 87.8 | 58.1 | 66.9 | 9.3 |
| QwenCoder-2.5-14B-LoRA | 78.0 | 50.9 | 66.2 | 5.3 |
| QwenCoder-2.5-14B-SFT | 79.9 | 42.4 | 36.8 | 3.1 |
| NextCoder-14B | 89.8 | 60.2 | 72.2 | 12.2 |
| QwenCoder-2.5-32B | **90.2** | 61.0 | 72.9 | 16.4 |
| QwenCoder-2.5-32B-LoRA | 82.3 | 52.4 | 60.2 | 6.7 |
| QwenCoder-2.5-32B-SFT | 81.7 | 49.5 | 66.9 | 8.4 |
| NextCoder-32B | 88.9 | **62.4** | **74.7** | **23.6** |

*Table 5.* Comparison of base QwenCoder-2.5 models of different sizes and their SeleKT-enhanced versions across three code editing benchmarks.

Table 5 demonstrates the performance of our SeleKT algorithm across various model sizes, including multilingual capabilities measured by the Aider Polyglot bench-

mark. For the smaller 3B model, NextCoder-3B shows significant improvements over the base model across most benchmarks, with a substantial gain on the CanItEdit benchmark (+5.3%). Notably, while the QwenCoder-2.5-3B-SFT achieves slightly better performance on HumanEvalFix, our approach excels on other benchmarks. At the 14B scale, NextCoder-14B consistently outperforms all baseline variants, achieving gains across all four benchmarks, with particularly impressive improvements on the Aider Polyglot benchmark (+2.9% over base model). For the largest 32B model, while there is a slight decrease in HumanEvalFix performance compared to the base model (-1.3%), NextCoder-32B achieves the highest scores across all other benchmarks, with a remarkable improvement on Aider Polyglot (+7.2%). These results demonstrate that our training approach provides consistent gains across model sizes.

### 5.4. Preserving Pre-learned Knowledge

| Models | HumanEval+ | MBPP+ |
|---|---|---|
| DeepSeekCoder-6.7B | 71.3 | 65.6 |
| DeepSeekCoder-6.7B-LoRA | 64.6 | 64.3 |
| DeepSeekCoder-6.7B-SFT | 70.1 | 59.5 |
| DeepSeekCoder-6.7B-TIES | 70.1 | 64.0 |
| DeepSeekCoder-6.7B-SeleKT | 73.2 | 65.3 |
| QwenCoder-2.5-7B | 85.4 | 72.5 |
| QwenCoder-2.5-7B-LoRA | 81.7 | 70.9 |
| QwenCoder-2.5-7B-SFT | 79.3 | 67.2 |
| QwenCoder-2.5-7B-TIES | 82.3 | 71.7 |
| NextCoder-7B | 84.8 | 72.0 |

*Table 6.* Comparing (% accuracy) of base and fine-tuned models on code generation benchmarks.

The results in Table 6 show that our SeleKT method largely preserves the code generation capabilities of the original models, as evaluated on the HumanEval+ and MBPP+ (Liu et al., 2023) benchmarks. For both DeepSeekCoder-6.7B and Qwen2.5-Coder-7B, full fine-tuning (SFT) and LoRA lead to performance drops of up to 6.7 accuracy points. In contrast, model merging using TIES helps mitigate these drops significantly. Notably, our method consistently outperforms TIES, further reducing performance degradation.

| Model | MMLU | GSM8K |
|---|---|---|
| QwenCoder-2.5-7B | 53.0 | 83.40 |
| QwenCoder-2.5-32B | 71.9 | 93.71 |
| NextCoder-7B | 54.5 | 81.65 |
| NextCoder-32B | 72.7 | 92.65 |

*Table 7.* Accuracy (%) on MMLU and GSM8K benchmarks.

The results in Table 7 demonstrate that NextCoder-32B improves on MMLU scores (+0.8 points), but has a small regression on GSM8K (-1.06 points). Overall, these results

demonstrate that our training methodology induces specialized code editing capabilities without sacrificing math problem solving and natural-language understanding abilities.

## 5.5. Effectiveness of Synthetic Data

Table 8 compares the performance of DeepSeekCoder-6.7B fine-tuned separately on CommitPackFT and on the synthetic data generated by our pipeline. While synthetic data offers a performance advantage over CommitPackFT, the latter represents a high-quality sample of real-world developer commits. To enhance generalizability, we incorporate both datasets in our fine-tuning process. Performance variations across different data sizes are further analyzed in Section A.2.

| Data | HumanEvalFix | CanItEdit | Aider |
|------|--------------|-----------|-------|
| CommitPackFT | 59.8 | 37.6 | 21.1 |
| Synthetic | **68.3** | **41.4** | **33.8** |

*Table 8.* Comparison of synthetic data and CommitPackFT data when used for fine-tuning the DeepSeekCoder-6.7B model.

## 5.6. Performance on Aider and Aider Polyglot Benchmarks

Evaluating code-editing models requires benchmarks that assess both general coding capabilities and multi-language proficiency. The Aider Polyglot benchmark has gained prominence for evaluating multilingual coding capabilities, featuring 225 exercises specifically selected as the most challenging problems from Exercism across multiple programming languages.

NextCoder models demonstrate impressive performance on both Aider and Aider Polyglot benchmarks compared to state-of-the-art code LMs (Figure 4). NextCoder-32B scores 74.7% on Aider, outperforming 71.4% of GPT-4o (2024-11-20) and approaching top models like Gemini-exp-1206 (80.5%). It also achieves 23.6% on Aider Polyglot against GPT-4o's 18.2%. Similarly, NextCoder-14B matches Deepseek-V2.5 on Aider (72.2%) despite having orders of magnitude fewer parameters. These results demonstrate that our training pipeline helps smaller, more efficient models to compete with much larger, more resource-intensive alternatives. The Aider Polyglot benchmark results further validate our approach, with all NextCoder models achieving relative performance advantages over comparably sized alternatives. While we demonstrate clear benefits in Aider Polyglot, the generally lower scores on this benchmark across all models reflect the benchmark's inherent difficulty with multi-language challenges. This suggests scope for improvement and line of investigation for our future work.

## 5.7. Ablation of Choices in SeleKT

**Effectiveness of Sparsity** To investigate the impact of sparsity parameter $\alpha$ in the SeleKT algorithm, we fine-tune the Qwen2.5-Coder-7B model with different $\alpha$ values: 0.05, 0.2, and 0.5. We fix the periodicity $M$ to the epoch boundary which is our default setting, and the number of epochs to 3. As seen in Table 9, with $\alpha = 0.05$ (most sparse, selecting only 5% of parameters to be updated), the model achieves the best overall performance. This is in agreement with our motivation that the updates remain tightly close to the base model in the $L_0$ sense to avoid overfitting, while selecting the parameters to be updated globally and periodically.

| $\alpha$ | HumanEvalFix | CanItEdit | Aider |
|------|--------------|-----------|-------|
| 0.05 | 81.1 | **50.5** | **65.7** |
| 0.2 | 76.8 | 45.7 | 53.4 |
| 0.5 | **81.7** | 43.3 | 54.9 |

*Table 9.* Ablation of the sparsity factor in the SeleKT algorithm used to fine-tune Qwen2.5-Coder-7B.

**Effectiveness of Periodicity** To examine the impact of the periodicity parameter $M$ in SeleKT, we fine-tune Qwen2.5-Coder-7B using different values of $M$ (0.1, 0.5, and 1.0 times the number of mini-batches in an epoch) while keeping the sparsity parameter $\alpha = 0.05$. Additionally, we compare against a baseline that follows full fine-tuning (without sparse updates) but applies a single SeleKT operation at the end, mimicking model merging techniques. As shown in Table A.6, aligning periodicity with epoch boundaries yields the best results for CanItEdit and Aider, whereas lower periodicity leads to worse performance. Performing a single SeleKT operation at the end, similar to model merging, achieves the high accuracy only on HumanEvalFix. These findings indicate that sparse updates are essential for robust fine-tuning, but excessive frequency may not be beneficial.

## 6. Conclusions

The ability of code LMs to accurately edit code at different scales and based on diverse instructions is central to their use in software engineering. In this paper, we present the next step in enhancing the code-editing ability of code LMs by developing a synthetic data pipeline and a robust adaptation algorithm SeleKT. Our pipeline produces diverse data that improves model performance and the adaptation algorithm ensures that the general, pre-learned abilities of the models are not lost during fine-tuning. We comprehensively evaluate our method on multiple code editing and generation benchmarks to establish these claims. In future, we want to extend our pipeline to cover more scenarios and evaluate SeleKT in tasks other than code-editing such as mathematical reasoning and natural language tasks.

## Impact Statement

This paper presents work whose goal is to advance the field of language models for code. We do not foresee any potential societal consequences related to the specific contributions of our work.

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

# A. Appendix

## A.1. Additional Details

### A.1.1. FINE-TUNING DATASET

| Language | Instances | Tokens(M) |
|----------|-----------|-----------|
| Python | 42094 | 9.50 |
| C | 6854 | 1.50 |
| C++ | 3816 | 0.90 |
| Java | 13443 | 3.25 |
| JS | 51704 | 11.60 |
| Rust | 2316 | 0.51 |
| Go | 4950 | 1.08 |
| Kotlin | 1819 | 0.44 |
| Total | 126996 | 28.78 |

*Table A.1.* Number of instances from CommitPackFT after filtration.

For CommitPackFT, we retain only those examples that contain both source and target code for the code-editing task (see Table A.1). The initial dataset consists of 153K instances across eight languages, and after filtration, 127K instances remain.

### A.1.2. TRAINING SETUP

**SFT** We finetuned DeepSeekCoder-6.7B model on 4xH100 GPUs and Qwen2.5-Coder-7B model on 8xH100 GPUs. For efficient memory management, we employed sample packing to a maximum sequence length of 8192 tokens for DeepSeekCoder-6.7B and 16384 tokens for Qwen2.5-Coder-7B, with batch sizes of 4 and 1 per GPU, respectively. Gradient accumulation steps were set to 4, resulting in respective effective batch sizes of 64 and 32. Additionally, DeepSeed's ZeRO Stage 3 (Rajbhandari et al., 2020) offloading to CPU, using bfloat16 for memory optimizations, was applied to both models.

**LoRA** The Qwen2.5-Coder-7B model was fine-tuned using LoRA (Hu et al., 2021), with a rank of 64 applied to all linear layers of the base model. The LoRA hyperparameters included an $\alpha$ value of 16 and a dropout rate of 0.05, without training the bias term. Batch size remained the same as in the SFT setup.

**SeleKT** The models were fine-tuned on 8xH100 GPUs. Sample packing, batch sizes and gradient accumulation steps were configured as reported for SFT. The samples were packed into a maximum sequence length of 8192 tokens for Deepseek and 16384 tokens for Qwen, with gradient accumulation steps of 4 and an effective batch size of 128 for Deepseek and 32 for Qwen. DeepSpeed's ZeRO Stage 2 offloading to CPU was used for both models with the bfloat16 data type for memory optimization. Additionally, the SeleKT algorithm was deployed with a sparsity factor of $\alpha = 0.05$ and a periodicity factor $M$ set to the epoch boundary.

**TIES** For the TIES variant of both DeepSeekCoder-6.7b and Qwen2.5-Coder-7b, we performed model merging by integrating the final third checkpoint into their respective base models. The merging process was conducted with both **density** and **weight** parameters set to 0.5, without applying normalization or int8 quantization.

### A.1.3. BENCHMARKS

We compared all the models on the following well-established benchmarks for code-editing (Table 3).

**CanItEdit** (Cassano et al., 2023) benchmark measures class and function-level code editing abilities of language models in Python for domains like Data Science, Mathematics, and Language Processing. Each problem instance is accompanied with descriptive (verbose) and lazy (terse) instructions, and correctness of edits is measured using execution accuracy over test cases. For this benchmark, we used a temperature of 0.2, top_p 0.95 and reported pass@1,1 scores.

**NoFunEval** (Singhal et al., 2024) benchmarks language models for their ability to edit file-level code in multiple programming languages based on non-functional requirements such as improving latency, resource utilization, security, and maintainability of existing code. Each problem instance is associated with four different types of prompts. Correctness of edits is

measured using run-time improvements, static-analysis based tools like CodeQL, or DiffBLEU scores, depending upon the non-functional requirement. We used greedy sampling for this benchmark and reported pass@1,1 scores.

**Aider** code-editing benchmark (Gauthier, 2024a) offers 133 small coding exercises in Python from Exercism dataset requiring an LM to edit python file for implementing a function or class as per natural language instructions. For this benchmark, we used a temperature of 0 and a whole-format setup (prompting the model to rewrite the entire code) and reported pass@2 scores.

**Aider Polyglot** code-editing benchmark (Gauthier, 2024b) offers 225 coding exercises from Exercism dataset. It contains exercises in multiple programming languages: C++, Go, Java, JavaScript, Python and Rust. The setup was the same as for Aider's code-editing benchmark. For this benchmark, we used a temperature of 0 and a whole-format setup (prompting the model to rewrite the entire code) and reported pass@2 scores.

**HumanEvalFix** benchmark (Muennighoff et al., 2023) evaluates models on the bug-fixing task, where models are given a code snippet along with an instruction to fix the code. We used a temperature of 0.2, top_p 0.95 and reported pass@1,1 scores for this task.

We also evaluate the models on standard code generation benchmarks.

**HumanEval+** benchmark (Liu et al., 2023) tests the models' performance in generating correct code based on textual descriptions with high-quality test cases. We used greedy sampling for this benchmark and reported pass@1,1 scores.

**MBPP+** benchmark (Liu et al., 2023) focuses on evaluating the models' ability to solve programming tasks that require mathematical reasoning and algorithmic thinking. We used greedy sampling for this benchmark and reported pass@1,1 scores.

Additionally, we evaluate the models on non-coding benchmarks.

**MMLU** benchmark (Hendrycks et al., 2021) (STEM subset) comprises 3.15K problems spanning Physics, Chemistry, Biology, Computer Science, Mathematics, and Engineering. We employed the few-shot setting (N=4) with the prompt configuration from Qwen models' official evaluation script.

**GSM8K** benchmark (Cobbe et al., 2021) contains grade-school mathematical problems that test reasoning abilities. For evaluation, we used the few-shot setting (N=4) with the prompt configuration from Qwen models' official evaluation script.

### A.1.4. HYPERPARAMETER TUNING FOR BASELINES

| Method | CanItEdit | HumanEvalFix | Aider |
|--------|-----------|--------------|-------|
| SFT | 46.2 | 78.6 | 52.6 |
| LoRA | 43.8 | 79.9 | 54.1 |
| SeleKT | **50.48** | **81.10** | **65.70** |

*Table A.2.* Results for the best versions of LoRA and SFT for Qwen2.5-Coder-7B

To ensure fair comparison between our proposed method and baseline approaches, we conducted systematic hyperparameter tuning experiments for SFT and LoRA methods. Since SeleKT was optimized with a learning rate of 1e-5 and weight decay of 0.0, we explored comparable configurations for the baseline methods. For both SFT and LoRA, we investigated learning rates of 2e-6 and 5e-6 combined with weight decay values of 0.10 and 0.05. For LoRA specifically, we further evaluated rank values ranging from 16 to 64 and alpha parameters of 8 and 16.

While this comprehensive tuning process yielded performance improvements for both baseline methods, Table A.2 demonstrates that a substantial performance gap persists between these optimized baselines and our NextCoder-7B model. Across all three evaluation benchmarks (CanIEdit, HumanEvalEdit, and Aider), SeleKT consistently outperforms the tuned baseline approaches. The most pronounced difference appears in the Aider benchmark, where SeleKT achieves a 11.6% improvement over the best baseline. These results underscore the effectiveness of our selective parameter updating approach beyond what can be achieved through standard hyperparameter optimization of conventional fine-tuning methods.

A.1.5. DECONTAMINATING SYNTHETIC TRAINING DATA

To ensure the integrity of our evaluations, we adopted the decontamination procedure used in prior works such as Star-Coder (Li et al., 2023; Lozhkov et al., 2024). Specifically, we applied near-duplicate detection using MinHash and Locality Sensitive Hashing (LSH) to identify and remove any potential overlaps between our training dataset and evaluation benchmarks (CanItEdit, HumanEvalFix, Aider, NoFunEval). Our decontamination analysis confirmed 0% data leakage, verifying that no benchmark data was present in the training set.

## A.2. Robustness of SeleKT across dataset sizes

| Dataset size | CanItEdit | HumanEvalFix | Aider |
|---|---|---|---|
| Base model (QwenCoder-2.5-7B) | 48.1 | 73.8 | 59.4 |
| 25% | 47.67 | 80.20 | 60.90 |
| 50% | 48.57 | **81.43** | 62.70 |
| 75% | 49.01 | 81.02 | 63.80 |
| 100% (NextCoder-7B) | **50.48** | 81.10 | **65.70** |

*Table A.3.* Performance of QwenCoder-2.5-7B when finetuned on splits of different sizes of our dataset

To assess scalability w.r.t. to training data size, we finetuned the QwenCoder-2.5-7B model on varying fractions (random sampling 25%, 50% and 75%) of our dataset which includes both synthetic and CommitPackFT data. The results are presented in the Table A.3. All models were trained for 3 epochs. The results show a clear trend: while performance on CanItEdit and Aider sees a drop at 25% w.r.t to the base model, increasing the dataset size consistently improves performance across all benchmarks (CanItEdit, HumanEvalFix, and Aider).

## A.3. Human Evaluation of Synthetic Dataset

We conducted a pilot human-study to assess the quality of the generated training dataset. In this study, we involved three participants who have 3-4 years of experience in software development with strong expertise in Python. We randomly selected 100 samples from the Python split of our synthetic dataset and asked participants to rate each sample on the scale of 1-5 (1 being poor quality and 5 being excellent quality) on the following three questions:

**i) Instruction Usefulness:** How well does the detailed instruction capture a potential code-editing scenario with respect to the original code?

**ii) Instruction Consistency:** How consistent are the three styles of instructions (detailed, concise and human-like) with each other and with the respective styles?

**iii) Solution Correctness:** How well does the edited code match the edit described in the detailed instruction?

| Metric | Participant 1 | Participant 2 | Participant 3 | Overall Mean | Overall SD |
|---|---|---|---|---|---|
| Instruction Usefulness | 4.92 ± 0.27 | 4.93 ± 0.26 | 4.38 ± 0.60 | 4.74 | 0.48 |
| Instruction Consistency | 4.29 ± 0.48 | 4.88 ± 0.32 | 4.55 ± 0.54 | 4.57 | 0.51 |
| Solution Correctness | 4.92 ± 0.27 | 4.96 ± 0.24 | 4.60 ± 0.51 | 4.83 | 0.40 |
| **Overall Mean ± SD** | **4.71 ± 0.46** | **4.92 ± 0.28** | **4.51 ± 0.56** | | |

*Table A.4.* Evaluation of Instruction Usefulness, Consistency, and Solution Correctness

In the table A.4, we present the mean (along with standard deviations) ratings by participant and by question. The scores are consistently close to the highest score of 5 across all participants and questions. This provides a strong indication of human-machine alignment, with low to moderate variance. This study helps validate that our synthetic data generation pipeline is able to generate samples that meet human expectations in terms of quality and consistency.

In the table A.5, we have shown the scores distribution. Clearly, all samples received scores 3 (neutral quality) or above on all the questions. We particularly inspected the samples that received the neutral rating (score 3) since those were perceived as relatively low-quality samples by one or more participants. We made the following observations:

| Score (Higher is better) | Instruction Usefulness | Instruction Consistency | Solution Correctness |
|---:|---:|---:|---:|
| 5 | 229 | 175 | 250 |
| 4 | 65 | 122 | 48 |
| 3 | 6 | 3 | 2 |
| **Total** | **300** | **300** | **300** |

*Table A.5.* Distribution of Evaluation Scores

**i) Instruction-Edit Misalignment:** In some cases, instructions correctly described the intent but the edits were not entirely appropriate. For example, in response to an instruction to handle datetime parsing, the edited code parsed dates against raw strings, which would cause runtime errors.

**ii) Incomplete Error Handling:** Some examples did introduce error handling, but overlooked edge cases (e.g., what if the 'tasks.json' file exists but is empty?).

**iii) Style Inconsistency:** A few participants noted that stylistic or structural variations across instruction formats led to minor misunderstandings of the code-editing intent.

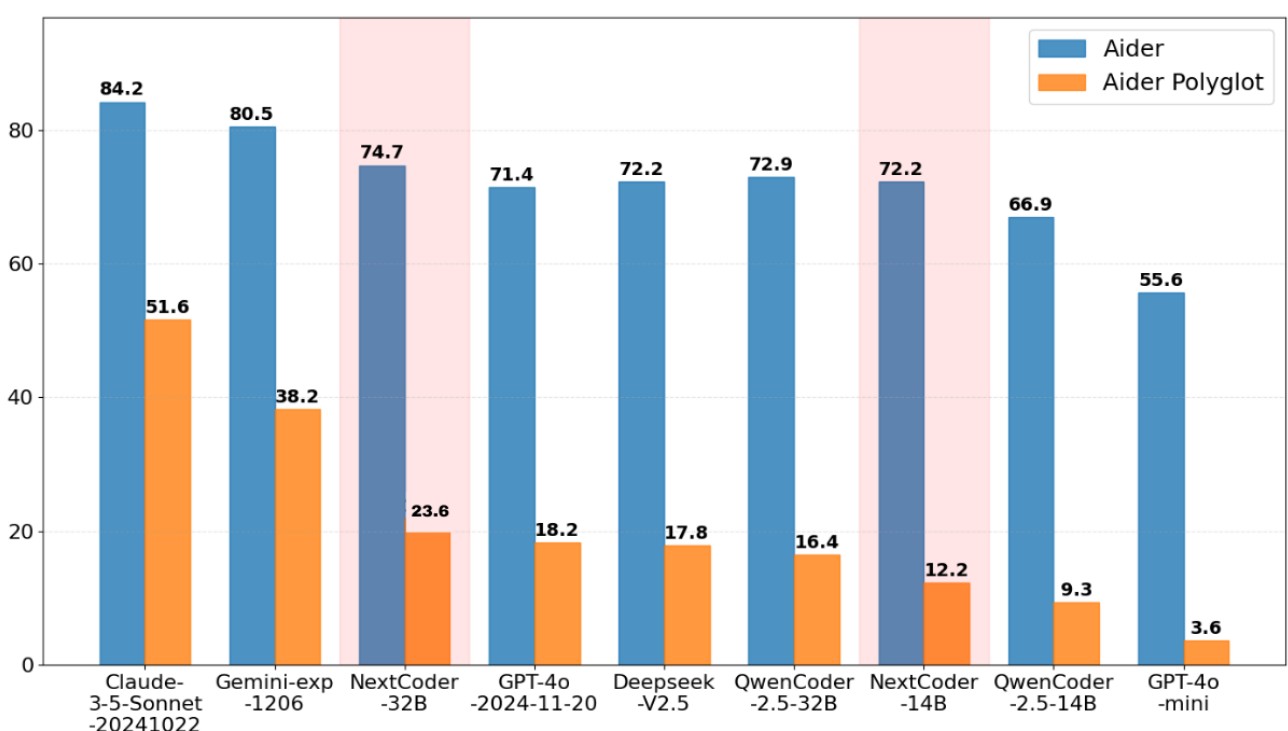

*Figure 4.* Performance of state-of-the-art code LMs on Aider and Aider Polyglot benchmarks. NextCoder-x is our code-editing model with Qwen2.5-Coder-x as the base, fine-tuned using the proposed SeleKT algorithm on synthetic and real code editing tasks. Baseline scores are sourced from the official leaderboard (Gauthier, 2024b).

| M | HumanEvalFix | CanItEdit | Aider |
|:---:|:---:|:---:|:---:|
| 0.1 Epoch | 80.5 | 37.1 | 51.1 |
| 0.5 Epoch | 83.5 | 50.4 | 59.4 |
| 1.0 Epoch | 81.1 | **50.5** | **65.7** |
| At the end | **84.2** | 50.0 | 53.2 |

*Table A.6.* Performance comparison of using SeleKT with different values of $M$ on the Qwen2.5-Coder-7B model.

## A.4. Prompts

You are an expert programmer tasked with creating a programming problem and corresponding code based on a given seed code. Your task is to understand the concepts and patterns in the seed code and create an inspired but different problem with intentionally imperfect code.

CONTEXT:
- Modular Level: {level} (This defines whether you should create a function-level, class-level, or file-level solution)
- Improvement Areas Needed:
  {area1_name}: {area1_desc}
  {area2_name}: {area2_desc}
  {area3_name}: {area3_desc}

 The solution should have deficiencies in these specified areas that can be improved later

Follow these steps:
1. Carefully analyze the given seed code to understand its core concepts and patterns.
2. Create a NEW programming problem that is inspired by these concepts but is not a direct copy.
3. Write a code solution that:
  - Matches the specified modular level
  - Takes inspiration from the seed code but creates a different implementation
  - Contains natural imperfections and inconsistencies
  - Implements core functionality but with flaws
  - May have unimplemented critical sections or clear bugs
  - Mixes different approaches to solving similar problems
  - Looks like code written by someone learning or in a hurry

The code should feel authentic - like something found in a real codebase with organic issues. Do not include comments about potential improvements or issues. The code should work for basic cases but have natural flaws in its implementation.

IMPORTANT NOTE: Do not add comments pointing out issues or suggesting improvements. The code should stand on its own with its natural imperfections.

Your output MUST strictly follow this format using the exact delimiters:

###PROBLEM_STATEMENT###
Write a clear, detailed problem statement describing what the code should accomplish.
The problem should be non-trivial and require a substantial solution.
###END_PROBLEM_STATEMENT###

###ORIGINAL_CODE###
Provide the code solution here, including comments.
The code should be functional but contain intentional deficiencies in the specified areas.
###END_ORIGINAL_CODE###

###METADATA###
MODULAR_LEVEL: {level}-level
LANGUAGE: {lang}

IMPROVEMENT_AREA_1: [name of the first improvement area]
LINES: [specific line numbers where improvements can be made, comma-separated or ranges e.g., 1,3,5-8]
DESCRIPTION: [detailed description of what deficiencies exist and how they could be improved]
TYPE: [specific type of issue within this improvement area]

IMPROVEMENT_AREA_2: [name of the second improvement area]
LINES: [affected line numbers]
DESCRIPTION: [detailed description of deficiencies]
TYPE: [specific type of issue]

IMPROVEMENT_AREA_3: [name of the third improvement area]
LINES: [affected line numbers]
DESCRIPTION: [detailed description of deficiencies]
TYPE: [specific type of issue]
###END_METADATA###

IMPORTANT:
- The code should be functional but intentionally suboptimal in the specified improvement areas
- Include comments in the code to help understand the logic
- Do not create trivial problems; ensure the solution has sufficient complexity
- Ensure deficiencies are realistic and improvable
- Follow the exact delimiter format - do not modify the delimiter strings
- Do not include any text outside the delimited sections

Here's the seed code to inspire your problem and solution:

```{lang}
{seed_code}
```

*Figure 5.* Prompt used for generating a problem and source code conditioned on the given seed code.

You are an expert programmer tasked with generating three different corrected versions of a code that has specific issues identified in the metadata. Based on the original improvement areas requested and the issues found, you will generate:
- An improved solution that implement the improvements

Solution should specifically address the improvements requested in the original improvement areas while fixing the issues identified in the metadata.

CONTEXT:
## Problem Statement:
{problem}

## Original Code:
```{lang}
{code}
```

## Requested Improvement Areas:
{area1_name}: {area1_desc}
{area2_name}: {area2_desc}
{area3_name}: {area3_desc}

## Identified Issues (Metadata):
{metadata}

Your task is to generate:
An improved solution that:
  - Fix all identified issues from metadata
  - Follow best coding practices
  - Implement proper error handling
  - Use efficient and maintainable approaches
  - Include clear comments explaining the improvements
  - Include all necessary imports
  - May use good implementation strategies

Your output MUST strictly follow this format using the exact delimiters:

###IMPROVED_SOLUTION_1###
# All imports here
[First version of the improved code with detailed comments explaining improvements]
###END_IMPROVED_SOLUTION_1###

###DIFFERENCES_EXPLAINED###
IMPROVED:
[Brief description of the approach and key improvements]

IMPORTANT:
- Solution must be fully functional
- Include descriptive comments explaining the implementation and improvements
- Solution should include ALL required imports
- Solution should be complete and standalone
- Maintain the same interface/API as the original code

*Figure 6.* Prompt used for generating the target code (improved code).

You are an expert prompt engineer tasked with generating three different types of instructions that guide an LLM to transform the original code into the improved version. Use the provided context to generate detailed, human-like, and conversational instructions.

CONTEXT:
## Problem Statement:
{problem}

## Original Code:
```{lang}
{code}
```

## Target Improved Version:
```{lang}
{edited_code}
```

## Key Improvements Made:
{explanations}
Generate four different instruction formats and Your output MUST strictly follow this format using the exact delimiters:

###DETAILED_INSTRUCTION###
[Generate a detailed instruction (not exceeding 8-10 lines) that:
- Clearly outlines each improvement needed
- Specifies exactly what changes are required
- Mentions specific functions/areas to modify
- Maintains clarity while being concise
Should provide enough detail for LLM to understand the required changes.]
###END_DETAILED_INSTRUCTION###

###CONCISE_INSTRUCTION###
[Generate a concise instruction (3-4 lines) that:
- Contains essential improvement points
- Covers all necessary changes
- Is clear but not overly detailed
Should provide just enough information to guide the changes.]
###END_CONCISE_INSTRUCTION###

###HUMAN_INSTRUCTION###
[Generate a very brief, human-like instruction that:
- Uses natural language
- Is concise (1-2 lines max)
- Captures core improvements needed
- Sounds like a quick dev chat message]
###END_HUMAN_INSTRUCTION###

###CONVERSATIONAL_INSTRUCTION###
[Generate a natural conversation between user and assistant that follows this flow:

USER: [General opening that naturally leads into code discussion]
ASSISTANT: [Engaging response following the user's direction]

USER: [Introduces the problem and initial code structure, showing interest in getting it right]
ASSISTANT: [Detailed acknowledgment and analysis of the problem/structure]

USER: Here's my current implementation: <code_placeholder> [Don't add the code just put the same place holder there]
ASSISTANT: [Provides clear explanation of code's current structure and functionality]

USER: [Final message containing:
- Moderate level of detail about required changes (between human and detailed)
- Clear instructions about what needs to be improved
- Reference to specific improvements needed
- Natural tone while being specific enough]]
ASSISTANT: [Brief acknowledgment of the requirements and indication that implementation will follow, followed by a
<code_output_placeholder>]
###END_CONVERSATIONAL_INSTRUCTION###

IMPORTANT:
- Each instruction type should guide towards the same end result
- Instructions should be clear and unambiguous
- Maintain natural language appropriate to each format
- Do not include actual code changes in the instructions
- Use the target improved version and key improvements as guide, but don't reference them directly in instructions
- Keep focus on what changes are needed, not how they were implemented

*Figure 7.* Prompt used for generating different types of instructions.

You are a quality assurance expert tasked with validating a training sample for code editing. Analyze the following components and provide a comprehensive assessment:

COMPONENTS TO ANALYZE:
## Seed Code:
```{lang}
{seed_code}
```
## Generated Data:
### Problem Statement: {problem}

### Original Code:
```{lang}
{original_code}
```
### Improved Edit:
```{lang}
{improved_edit}
```
### Instructions:
- Detailed: {detailed_instruction}
- Human: {human_instruction}
- Conversational:
{conversational_instruction}

Perform the following quality checks and provide scores (0-10) with explanations and Your output MUST strictly follow this format using the exact delimiters::
###COHERENCE_CHECK###
1. Original-Edit Alignment:
  - Do edits properly address the code's issues?
  - Are improvements meaningful and substantial?
  - Do changes align with requested improvement areas?
Score: [0-10]
Explanation: [Brief analysis]

2. Edit-Instructions Alignment:
  - Do instructions clearly guide towards the implemented changes?
  - Are all significant changes covered in instructions?
  - Is the instruction complexity appropriate for each format?
Score: [0-10]
Explanation: [Brief analysis]
###END_COHERENCE_CHECK###

###QUALITY_CHECK###
1. Code Quality:
  - Original code deficiencies: Are they realistic and fixable?
  - Edit improvements: Are they meaningful and well-implemented?
  - Code structure: Is it clear and maintainable?
Score: [0-10]
Explanation: [Brief analysis]

2. Instruction Quality:
  - Detailed: Clear, specific, and comprehensive?
  - Human: Natural, concise, and effective?
  - Conversational: Logical flow and clear final request?
Score: [0-10]
Explanation: [Brief analysis]

3. Training Value:
  - Will this help SLM learn code editing?
  - Are the examples diverse and meaningful?
  - Is complexity appropriate for training?
Score: [0-10]
Explanation: [Brief analysis]
###END_QUALITY_CHECK###

###FINAL_VERDICT###
Strengths:
- [List key strengths]
Weaknesses:
- [List areas needing improvement]
Recommendations:
- [Specific suggestions if any improvements needed]
###END_FINAL_VERDICT###

*Figure 8.* Prompt used for assessing the quality of the data generated.

## B. Instance Generated from Our Data Pipeline

We present a synthetic data instance generated using our pipeline in Figure 9.

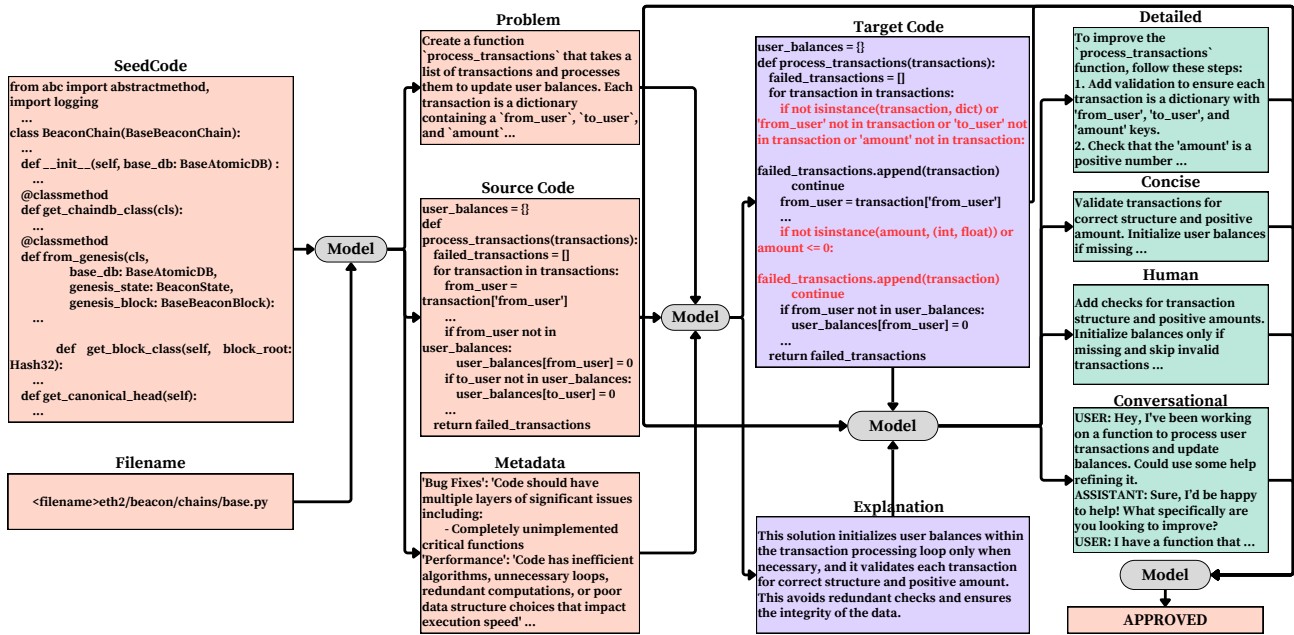

*Figure 9.* Figure showing an example passed through the data generation pipeline for GPT-4o.

# C. Examples Showing the Quality of Our Model Outputs

**Instruction:** Your task is to add the mine counts to empty squares in a completed Minesweeper board.
The board itself is a rectangle composed of squares that are either empty (`' '`) or a mine (`'*'`).

For each empty square, count the number of mines adjacent to it (horizontally, vertically, diagonally).
If the empty square has no adjacent mines, leave it empty.
Otherwise replace it with the adjacent mines count.

For example, you may receive a 5 x 4 board like this (empty spaces are represented here with the '·' character for display on screen):

```text
·*·*·
··*··
··*··
·····
```

Which your code should transform into this:

```text
1*3*1
13*31
·2*2·
·111·
```

**Input Code:**
```
def annotate(minefield):
    if not minefield or not all(len(row) == len(minefield[0]) for row in minefield):
        raise ValueError("The board is invalid with current input.")

    def count_mines(x, y):
        count = 0
        for i in range(max(0, x-1), min(x+2, len(minefield))):
            for j in range(max(0, y-1), min(y+2, len(minefield[0]))):
                if minefield[i][j] == '*':
                    count += 1
        return count

    result = []
    for i, row in enumerate(minefield):
        new_row = []
        for j, cell in enumerate(row):
            if cell not in ['*', '.']:
                raise ValueError("The board contains invalid characters.")
            if cell == '*':
                new_row.append('*')
            else:
                mine_count = count_mines(i, j)
                new_row.append(str(mine_count) if mine_count > 0 else ' ')
        result.append(''.join(new_row))

    return result
```

**Qwen Output:**

```
def annotate(minefield):
    if not minefield or not all(len(row) == len(minefield[0]) for row in minefield):
        raise ValueError("The board is invalid with current input.")

    def count_mines(x, y):
        ...

    result = []
    for i, row in enumerate(minefield):
        new_row = []
        for j, cell in enumerate(row):
            if cell == '*':
                new_row.append('*')
            else:
                mine_count = count_mines(i, j)
                new_row.append(str(mine_count) if mine_count > 0 else ' ')
        result.append(''.join(new_row))

    return result
```

**NextCoder Output:**

```
def annotate(minefield):
    if not minefield:
        return []

    # Validate the input board
    row_length = len(minefield[0])
    for row in minefield:
        if len(row) != row_length:
            raise ValueError("The board is invalid with current input.")
        for char in row:
            if char not in ' *':
                raise ValueError("The board is invalid with current input.")

    # Function to count mines around a given cell
    def count_mines(x, y):
        ...

    # Create a new board to store the annotated values
    annotated_board = []
    for i in range(len(minefield)):
        new_row = []
        for j in range(len(minefield[0])):
            if minefield[i][j] == '*':
                new_row.append('*')
            else:
                mine_count = count_mines(i, j)
                new_row.append(str(mine_count) if mine_count > 0 else ' ')
        annotated_board.append(''.join(new_row))

    return annotated_board
```

*Figure 11.* Example code-editing task (top) from Aider benchmark. NextCoder (RHS) solves the task correctly in the first attempt: The input can only contain spaces or '*'s. However, in the example in the prompt, '.' is used to represent space and the same is explicitly stated. Despite this, Qwen2.5-Coder-7B accepts inputs which contain characters other than '*' and space. This problem is fixed by NextCoder by rejecting any input which violates this constraint. The relevant code in NextCoder's output is highlighted in bold.

**Instruction:** In word search puzzles you get a square of letters and have to find specific words in them.
For example:
```text
jefblpepre
camdcimgtc
oivokprjsm
pbwasqroua
rixilelhrs
wolcqlirpc
screeaumgr
alxhpburyi
jalaycalmp
clojurermt
```
There are several programming languages hidden in the above square.
Words can be hidden in all kinds of directions: left-to-right, right-to-left, vertical and diagonal.
Given a puzzle and a list of words return the location of the first and last letter of each word.

**Input Code:**
```
class Point:
    def __init__(self, x, y):
        self.x = x
        self.y = y

    def __eq__(self, other):
        return self.x == other.x and self.y == other.y

class WordSearch:
    def __init__(self, puzzle):
        self.puzzle = puzzle
        self.rows = len(puzzle)
        self.cols = len(puzzle[0])

    def search(self, word):
        directions = [
            (0, 1),  # left-to-right
            (0, -1), # right-to-left
            (1, 0),  # top-to-bottom
            (-1, 0), # bottom-to-top
            (1, 1),  # top-left to bottom-right
            (-1, -1),# bottom-right to top-left
            (1, -1), # top-right to bottom-left
            (-1, 1)  # bottom-left to top-right
        ]

        for row in range(self.rows):
            for col in range(self.cols):
                for dx, dy in directions:
                    if self._search_from(row, col, word, dx, dy):
                        return (Point(col, row), Point(col + (len(word) - 1) * dx, row + (len(word) - 1) * dy))
        return None

    def _search_from(self, row, col, word, dx, dy):
        for i in range(len(word)):
            if row < 0 or row >= self.rows or col < 0 or col >= self.cols:
                return False
            if self.puzzle[row][col] != word[i]:
                return False
            row += dx
            col += dy
        return True
```

**Qwen Output:**
```
class Point:
    ...
class WordSearch:
    def __init__(self, puzzle):
        ...

    def search(self, word):
        ...

        for row in range(self.rows):
            for col in range(self.cols):
                for dx, dy in directions:
                    if self._search_from(row, col, word, dx, dy):
                        return (Point(col, row), Point(col + (len(word) - 1) * dx, row + (len(word) - 1) * dy))
        return None

    def _search_from(self, row, col, word, dx, dy):
        for i in range(len(word)):
            if row < 0 or row >= self.rows or col < 0 or col >= self.cols:
                return False
            if self.puzzle[row][col] != word[i]:
                return False
            row += dx
            col += dy
        return True
```

**NextCoder Output:**
```
class Point:
    ...

class WordSearch:
    def __init__(self, puzzle):
        ...

    def search(self, word):
        ...
        for direction in directions:
            for i in range(self.rows):
                for j in range(self.cols):
                    if self._search_from_point(word, i, j, direction):
                        return (Point(j, i), Point(j + (len(word) - 1) * direction[0], i + (len(word) - 1) * direction[1]))
        return None

    def _search_from_point(self, word, start_row, start_col, direction):
        for k in range(len(word)):
            row = start_row + k * direction[1]
            col = start_col + k * direction[0]
            if row < 0 or row >= self.rows or col < 0 or col >= self.cols or self.puzzle[row][col] != word[k]:
                return False
        return True
```

*Figure 10.* Example code-editing task (top) from Aider benchmark. NextCoder (RHS) solves the task correctly in the first attempt, while Qwen2.5-Coder-7B (LHS) gets it wrong in both the attempts: The directions are represented by pairs of the form (a,b) where a is the y-coordinate and b is the x-coordinate. However, Qwen2.5-Coder-7B incorrectly assumes that a and b are respectively x and y coordinates and generates edited code accordingly. NextCoder interprets the instruction accurately and generates correct code. The relevant code in NextCoder's output is highlighted in bold.

