# OpenReview forum: "NextCoder: Robust Adaptation of Code LMs to Diverse Code Edits"
_ICML.cc/2025/Conference — ICML 2025 poster_

### Official Review · Reviewer_itGF · 2025-03-09

**Overall Recommendation:** 3

**Summary:**

This paper addresses two issues: (1) enhancing code language models on code-editing tasks; and (2) mitigating catastrophic forgetting caused by task-specific fine-tuning. To address (1), it proposes a method for synthesizing high-quality code-editing data; to address (2), it introduces Selective Knowledge Transfer (SeleKT), a low-rank optimization technique that limits parameter updates. Experiments on four code-editing benchmarks demonstrate that the approach achieves state-of-the-art performance with models of comparable scale while preserving robust code generation capabilities.

**Claims And Evidence:**

The claims "Existing code LMs are deficient in handling diverse code-editing tasks" and "suppressing catastrophic forgetting in general models after fine-tuning" raise concerns about the real-world applicability of research in the field of code language models.

While the authors criticize the quality of existing code editing dataset, my concern lies in why the LLM-generated dataset is realistic and representative? Specifically, the real-world code edits for a change-requirement (a.k.a, a commit) can distributed across many files in a code repository, how could the synthesized dataset to represent such distribution.

**Essential References Not Discussed:**

NA

**Experimental Designs Or Analyses:**

Lack of evaluation of the generated benchmark, why it is useful for training a practical code editing model. I suggest the trained model shall be evaluated on some real-world commit dataset as collected in [2].

[2] Chenyan Liu, Yufan Cai, Yun Lin, Yuhuan Huang, Yunrui Pei, Bo Jiang, Ping Yang, Jin Song Dong, and Hong Mei. CoEdPilot: Recommending Code Edits with Learned Prior Edit Relevance, Project-wise Awareness, and Interactive Nature (ISSTA 2024)

**Methods And Evaluation Criteria:**

Code editing can involve both edit location and edit generation, as in [2]. The authors seems to miss the edit generators such as

[1] Saikat Chakraborty, Yangruibo Ding, Miltiadis Allamanis, and Baishakhi Ray. 2022. CODIT: Code Editing With Tree-Based Neural Models. IEEE Transactions on Software Engineering 48, 4 (2022), 1385–1399. https://doi.org/10.1109/TSE. 2020.3020502

[2] Chenyan Liu, Yufan Cai, Yun Lin, Yuhuan Huang, Yunrui Pei, Bo Jiang, Ping Yang, Jin Song Dong, and Hong Mei. CoEdPilot: Recommending Code Edits with Learned Prior Edit Relevance, Project-wise Awareness, and Interactive Nature (ISSTA 2024)

**Other Comments Or Suggestions:**

NA

**Other Strengths And Weaknesses:**

NA

**Questions For Authors:**

1. How do you evaluate the usefulness of the generated training datasets?
2. Whether your approach can address the repository-level code editing, which is more practical in the real world?

**Relation To Broader Scientific Literature:**

The topic of the paper is practical and useful.

**Theoretical Claims:**

NA

---

> ### Author Rebuttal · Authors · 2025-04-01
>
> We thank the reviewer for the insightful feedback on our work. We answer the questions asked by the reviewer below:
>
> > My concern lies in why the LLM-generated dataset is realistic and representative? Specifically, ..., how could the synthesized dataset to represent such distribution.
>
> Our synthetic data generation, by design, supports multiple correlated edits within a single task. Indeed, our pipeline generates multi-edit examples and we will add concrete examples in the appendix in the revised version. This approach mirrors real-world software development, where changes often span multiple classes and interrelated code segments. By generating data that captures the complexity of correlated changes, we ensure an authentic representation of complex code editing scenarios.
>
> To further validate our dataset's ability to handle multi-file edits, we finetuned larger QwenCoder-2.5 variants and evaluated them on [Aider Polyglot](https://aider.chat/2024/12/21/polyglot.html#the-polyglot-benchmark), a challenging benchmark with multi-file edit problems. Due to char limits, we are unable to give a concrete example in this response. However, we would be happy to share it if the reviewer wants, in another response. Our NextCoder models show significant performance gains, as summarized in the tables below.
>
> | **14B Models** | **Polyglot** |
> |:----------|:------------:|
> | QwenCoder-2.5-14B | 9.3 |
> | QwenCoder-2.5-14B-LoRA | 5.3 |
> | QwenCoder-2.5-14B-SFT | 3.1 |
> | **NextCoder-14B** | **12.2** |
>
> | **32B Models** | **Polyglot** |
> |:----------|:------------:|
> | QwenCoder-2.5-32B | 16.4 |
> | QwenCoder-2.5-32B-LoRA | 6.7 |
> | QwenCoder-2.5-32B-SFT | 8.4 |
> | **NextCoder-32B** | **21.9** |
>
> > I suggest the trained model shall be evaluated on some real-world commit datasets like [CoEdPilot](https://arxiv.org/abs/2408.01733).
>
> We agree that evaluation on real-world commits is essential -- and we have already done this. A subset of the NoFunEval benchmark [CoLM 2024], which includes five splits assessing a model’s ability to improve code on multiple non-functional requirements (e.g., runtime, safety, etc.), is derived from real-world commits in Android repositories. In particular, the latency and resource-utilization splits are derived from real commits. The results of these evaluations are discussed in Section 5.2 of our paper.
>
> Regarding CoEdPilot, while it represents an interesting benchmark derived from real-world commits, its focus on edit propagation (predicting edits across multiple locations based on patterns of previous edits) differs substantially from our work's objectives. Our approach aims to perform targeted code edits based on natural language instructions, rather than propagating edits across a codebase. Additionally, CoEdPilot's emphasis on fine-grained edit detection and classification at the line level (keep/insert/replace) wouldn't align well with our instruction-following paradigm. The fundamental difference is that our model requires explicit natural language instructions, whereas CoEdPilot infers edit patterns automatically. If required, we will manually write instructions for some of the instances from CoEdPilot and include results in the revised version.
>
> > Code editing can involve both edit location and edit generation, as in [CoEdPilot](https://arxiv.org/abs/2408.01733). The authors seems to miss the edit generators such as [CODIT](https://arxiv.org/abs/1810.00314).
>
> We thank the reviewer for pointing out CoEdPilot and CODIT; however, these methods target distinct scenarios. CoEdPilot focuses on edit propagation, predicting future edits based on past edits, and employs a fine-grained edit detection mechanism (Edit-propagating Line Locator) to classify edit types (keep/insert/replace) at the line level. In contrast, our approach finetunes a code LM to follow natural-language instructions for editing a given codebase, without requiring past edits or explicit edit localization. CODIT, on the other hand, predicts repetitive edits using a tree-based neural machine translation model, focusing on structured edit patterns rather than general-purpose instruction-following for code modifications. Additionally, CODIT predates modern LLM-based approaches, making its methodology different from ours. These distinctions clarify that our work does not overlook edit generators but instead addresses a separate formulation of the code-editing problem. We will incorporate these clarifications, comparing against CoEdPilot and CODIT, into the paper.
>
> > How do you evaluate the usefulness of the generated synthetic data?
>
> In Section 5.4, we provide empirical evidence demonstrating the utility of our synthetic data compared to traditional commit-based datasets. Specifically, we observe that fine-tuning DeepSeek-6.7B on our synthetic data yields a performance advantage over CommitPackFT (a filtered dataset of high-quality commit data from GitHub).

---

> > ### Comment · Reviewer_itGF · 2025-04-03
> >
> > I thank the authors' clarification, which largely addresses my concern. In this case, I would like to vote for this submission with weak acceptance.

---

> > > ### Author Response · Authors · 2025-04-07
> > >
> > > Thank you for giving due consideration to our response and increasing your score!

---

### Official Review · Reviewer_MF2X · 2025-03-10

**Overall Recommendation:** 4

**Summary:**

The authors present a comprehensive approach to enhance the code editing capabilities of language models while maintaining their pre-existing abilities. Their work addresses two fundamental challenges in this domain: the scarcity of high-quality fine-tuning data for code editing tasks and the phenomenon of catastrophic forgetting during domain adaptation.

The primary contributions of this research are threefold. First, the authors introduce a synthetic data generation pipeline to diverse code editing examples across eight programming languages. This pipeline systematically generates original code, modified code, and corresponding natural language instructions with varying levels of verbosity and styles, encompassing multiple edit types and code granularities.

Second, the authors propose the Selective Knowledge Transfer (SeleKT) algorithm, a robust adaptation method that employs dense gradient-based steps to identify critical weights for code editing tasks, followed by sparse projections onto the base model to mitigate overfitting. Unlike conventional approaches that predetermine updatable weights, SeleKT dynamically reassesses weight importance throughout the fine-tuning process based on magnitude changes.

Third, through empirical evaluation on four established benchmarks (CanItEdit, HumanEvalFix, NoFunEval, and Aider), the authors demonstrate that their adaptation of Qwen2.5-Coder-7B, named NextCoder, surpasses comparable models and even competes with substantially larger models on several tasks. Additionally, their experiments confirm that the SeleKT approach generalizes across model families, as evidenced by improvements in DeepSeekCoder-6.7B performance.

## update after rebuttal
I thank the authors for their detailed response and my fellow reviewers for their insightful comments. Based on the additional ablation results, I have revised my score to 4.

**Claims And Evidence:**

Most of the claims made in the paper are generally supported by convincing empirical evidence including

- NextCoder (using SeleKT) outperforms comparable models and even larger ones, on diverse code-editing benchmarks.
- Proposed Automatic Synthetic data pipeline yields gains over CommitPackFT data baseline.
0 SeleKT preserves pre-learned abilities is validated by Table 5, which shows that models fine-tuned with SeleKT retain their code generation capabilities better than those fine-tuned with alternative methods.

However, there are several claims which lack robust evidence. In particular

- Synthetic Data quality and generalization: The claim that the synthetic data pipeline captures real-world diversity is less substantiated, as the analysis relies heavily on select benchmark performance. We know that one can improve task performance by obtaining synthetic samples similar to the end task distributions. Since authors have not done any analysis on generating data overlap with benchmark data, it’s even not clear if they are inadvertently overfitting to test. Further, the proposed pipeline solely relies on LLM based automatic checks and efficacy of such checks is not evaluated by the authors.

- Lack of hyperparam tuning for baselines: While authors present extensive analysis for their own proposed algo SeleKT, I find the SFT, LoRA baseline numbers to be less convincing as authors show that SFT leads to huge degradation (65.6-> 59.5 on MBPP+). It’s not clear for how many steps the model was trained and what hyperpamas authors tried to reduce potential overfitting.

- Lack of data scaling curve: On synthetic data, authors do not show how scaling up data size improves performance and when do we see some sort of performance saturation.

-  SeleKT generalization: I strongly suggest authors to measure generalization beyond Humaneval and MBPP. For example, they can also use MMLU or some non-code related dataset to truly measure degradation in model quality.

**Essential References Not Discussed:**

While one can always cite additional papers, I think authors have cited relevant literature to connect paper to existing ideas.

**Experimental Designs Or Analyses:**

- Benchmark Selection and Evaluation: Authors used standard benchmarks for code edits (HumanEvalFix, Aider) as well as generic code generation performance (Humaneval, MBPP), which is a very reasonable choice. To measure if model retain performance on generic tasks, I would also preferred to include benchmarks beyond just code generation.
- Quality and Representativeness of synthetic data: The soundness of the synthetic data generation is critical. While the pipeline is innovative, the quality of the synthetic examples depends on the underlying LLMs used for generation, which may introduce systematic biases or fail to capture rare code-editing scenarios.
- Ablation Studies on Hyperparameters: For SeleKT algo, authors provide detailed ablations on key hyperparameters, such as the sparsity factor $alpha$. However, for baseline methods, authors don't provide details on hyperparam selection.
- Ablations on synthetic data: While the proposed data generation pipeline is reasonable, paper lacks ablations related to quality of generated data beyond end task performance, how scaling up data improves performance.

**Methods And Evaluation Criteria:**

The proposed methods and evaluation criteria in the paper are well-tailored to the problem of adapting code language models for diverse code edits.
- Synthetic Data Generation Pipeline: The approach builds on previous successes in using synthetic data for instruction tuning (e.g., self-instruct methods in natural language processing) and extends those ideas to the code domain, addressing the limitations of using only mined commit data. The observed gains on multiple benchmarks (see Tables 1 and 6) validate the effectiveness of the proposed synthetic data approach.
- Diverse Benchmark Datasets: The use of benchmarks such as CanItEdit, HumanEvalFix, Aider, and NoFunEval is appropriate because they capture a variety of editing scenarios—from function-level bug fixes to full file improvements and even non-functional aspects like performance or security. This diversity is crucial for assessing how well the adapted model handles the breadth of real-world code-editing tasks. Additionally, evaluation on generation benchmarks like HumanEval+ and MBPP+ ensures that any gains in code editing do not come at the expense of the model’s fundamental code generation and comprehension abilities.

**Other Comments Or Suggestions:**

I don't have other comments

**Other Strengths And Weaknesses:**

- Lack of baseline optimization: It is unclear if hyperparameter settings have been optimized for baseline used in paper including SFT, PEFT. The effectiveness of these comparisons hinges on whether each baseline was optimally tuned.
- Lack of data ablations on synthetic data quality: Authors should consider conducting scaling experiments to show how quantity of data improves performance and should conduct human annotations of a subsample to explain the quality and limitations of generated data.

**Questions For Authors:**

- Could you please provide SFT and LoRA fine-tuning details including hyperparams tried to reduce overfitting?
- To measure model's ability to retain generic performance, can you also evaluate it on other tasks such as MMLU?

**Relation To Broader Scientific Literature:**

- The idea of generating synthetic instruction-response pairs for fine-tuning has become a standard method in natural language processing, as seen in self-instruct pipelines (e.g., Wei et al., 2024a; Wang et al., 2023). The paper extends these ideas to code editing, building on methods like CodeAlpaca and Self-Instruct but specifically targeting the diversity of code modifications. Overall, it's a well-tested recipe to improve performance.
- The challenge of catastrophic forgetting in fine-tuning has been extensively studied (Goodfellow et al., 2013; Kirkpatrick et al., 2017). SeleKT is motivated by these works, aiming to strike a balance between acquiring new task-specific knowledge and retaining general capabilities. Further, techniques such as LoRA (Hu et al., 2021) and recent sparse adaptation methods (Nguyen et al., 2024b) update only a subset of parameters to avoid overfitting. Unlike these methods, which select parameters a priori, SeleKT dynamically reassesses which parameters to update by computing the full gradient periodically, then selecting the top-k updates. This approach is more adaptive and echoes ideas from model merging techniques like TIES (Yadav et al., 2024) but integrates the selection process during training rather than post-hoc.

**Theoretical Claims:**

The proof for Lemma 1 is correct in its intended scope—it shows that the selective update mechanism guarantees that at most $c$
parameters are altered relative to the base model. However, it is limited to a counting argument and does not offer any theoretical insights into the optimization dynamics or convergence behavior in non-convex settings.

---

> ### Author Rebuttal · Authors · 2025-04-01
>
> We thank the reviewer for the insightful feedback on our work. We answer the questions asked by the reviewer below:
>
> > Authors have not done any analysis on generating data overlap with benchmark ...
>
> The benchmarks considered in the paper are based on manually-created coding problems and solutions. Whereas, we used the training split of the StarCoder dataset, which is derived from GitHub commits, as our seed data. The synthetic data pipeline generates samples inspired by the seed data. This reduces the likelihood of unintended overlap with the benchmark data and prevents inadvertent overfitting.
>
> We agree on the importance of this point with the reviewer and quantitatively analyze overlap using the standard approach of decontamination used in StarCoder: [bigcode-dataset/decontamination](https://github.com/bigcode-project/bigcode-dataset/tree/main/decontamination). The decontamination report confirms 0% overlap between our training data and benchmarks.
>
> We will include a discussion on this and report the decontamination measurement in the revised version.
>
> > Lack of hyperparam tuning for baselines ...
>
> We have documented the specifics of our SFT and LoRA hyperparameters in Appendix A.1.2. For LoRA, we utilized the hyperparameters from the Qwen model official implementation [scripts](https://github.com/QwenLM/Qwen/blob/main/finetune.py#L55). To address the reviewer's concern, we have conducted an initial experiment on hyperparameter tuning for baseline methods. Given that SeleKT was trained with a learning rate of 1e-5 and weight decay of 0.0, we explored closer configurations for LoRA and SFT, testing learning rates of 2e-6 and 5e-6 with weight decay values of 0.10 and 0.05. Additionally, for LoRA, we experimented with ranks from 16 to 64 and alpha values of 8 and 16. While this tuning led to some improvements in both SFT and LoRA, a significant performance gap remains between these baselines and NextCoder-7B. We will include these updated results in the final version.
>
> > Not clear for how many steps the model was trained.
>
> We trained all our models for 3 epochs (Section 5.1).
>
> > Theoretical insights into the optimization dynamics or convergence behavior in non-convex settings
>
> Please refer to a detailed response to reviewer 3QHQ (3rd point).
>
> > Lack of data scaling curve
>
> Please refer to a detailed response to reviewer 3QHQ (2nd point).
>
> > To measure model's ability to retain generic performance, can you also evaluate it on other tasks such as MMLU?
>
> In addition to (a) the additional experiment on MMLU as suggested by the reviewer, we also conducted (b) experiments on GSM8K to further demonstrate the ability of our method to retain generic performance.
>
> For evaluation, we followed the few-shot setting (N=4) and the same prompt used in the Qwen models' official evaluation script. Given that our model is designed for code-related tasks, we focused on the STEM subset of MMLU, which contains 3.15K problems covering topics such as: Physics, Chemistry, Biology, Computer Science, Mathematics and Engineering. This subset aligns closely with the problem-solving and computational reasoning abilities expected from a code-editing model, making it a more meaningful evaluation of whether fine-tuning on code has impacted general problem-solving performance. For GSM8K, we considered the full benchmark.
>
>
> | Model                      | MMLU | GSM8K |
> | -------------------------- | -------- | -------- |
> | Qwen2.5-Coder-7B-Instruct  | 53.0     | 83.40 |
> | Qwen2.5-Coder-32B-Instruct | 71.9     | 93.71 |
> | NextCoder-7B      | 54.5     | 81.65 |
> | NextCoder-32B     | 72.7    | 92.65 |
>
> The above table presents the accuracy scores for our NextCoder models alongside the Qwen2.5-Coder models. These results substantiate the robustness of our approach, in particular the absence of catastrophic forgetting is evident. We will add these results to the paper.
>
> > Underlying LLM for data generation might introduce systematic biases or fail to capture rare-code editing scenarios.
>
> The reviewer raises important issues which can affect effectiveness of synthetic data generation. To mitigate these issues, we implemented a multi-faceted approach to ensure diversity and reduce systematic biases. Our synthetic data generation strategy relies on a diverse set of seed data (which, for example, is significantly larger than [WizardCoder’s](https://arxiv.org/pdf/2306.08568) 20K instances) as the basis (Section 3, point i), ensuring a broad initial representation of code editing contexts. We further enhance the coverage/diversity of our generated scenarios by incorporating three randomly selected improvement areas (Section 3, point i) for each synthetic data instance. This approach helps prevent the model from converging on a narrow set of editing patterns.
>
> By deliberately introducing randomness through multiple improvement areas and using a diverse initial dataset, we have tried to mitigate bias and coverage concerns.

---

### Official Review · Reviewer_UeAD · 2025-03-12

**Overall Recommendation:** 3

**Summary:**

This paper proposes an approach to handling diverse code-editing requirements. First, it introduces a synthetic data generation pipeline that begins with seed code samples and applies various editing criteria to produce high-quality training data. This pipeline generates pairs of original and modified code along with natural language instructions in different styles and verbosity levels. Second, the paper presents SeleKT, a model adaptation algorithm that identifies the most crucial weights for code editing using a dense gradient-based step, followed by a sparse projection onto the base model to prevent overfitting. Experimental results show that the resulting model, NextCoder, achieves strong performance across multiple code-editing benchmarks, surpassing comparably sized models and even outperforming some larger models in code-editing tasks.

## update after rebuttal
As the authors present well-designed research questions for studying human-machine alignment, supported by evidence-based performance improvements, the reviewer agrees that the proposed SeleKT methodology is both effective and practical in code-editing scenarios. Therefore, the reviewer raises the score.

**Claims And Evidence:**

- While the challenges discussed in the paper appear relevant to the code editing task, the reviewer is concerned that these issues are common across all machine learning tasks. The scarcity of high-quality fine-tuning data and the risk of catastrophic forgetting during fine-tuning are well-known problems in ML models in general. This raises concerns that the paper’s contribution may not be sufficiently distinct.

**Essential References Not Discussed:**

- No comment

**Experimental Designs Or Analyses:**

- The reviewer has examined the result analysis section and agrees that the proposed SeleKT method enhances code editing efficiency and delivers better performance. Additionally, the reviewer considers SeleKT a general methodology applicable to training open-source ANN models for improved code editing.

**Methods And Evaluation Criteria:**

- The reviewer has a particular interest in the quality of the generated dataset. In Line 210 (L), the authors state that they perform quality-based filtering by prompting the LLM to select high-quality examples. However, the reviewer believes that this step requires some level of human intervention, including: (1) Defining the criteria for assigning a specific score. (2) Providing demonstration examples to guide the LLM in evaluating dataset quality. (3) (If possible) Conducting an empirical study to assess whether the LLM's evaluation standards align with those of human experts. For examples of empirical study, please refer to section 4.4 in [1].

[1] AutoDSL: Automated domain-specific language design for structural representation of procedures with constraints, ACL’24

**Other Comments Or Suggestions:**

- [Table 3] There is excessive blank space below the table caption. Please check the typeset configuration.
- [line 337] The model name “DeepSeek-R1-Qwen-7B” is missing the \textsf formatting.

**Other Strengths And Weaknesses:**

- No comment.

**Questions For Authors:**

- No question.

**Relation To Broader Scientific Literature:**

- The paper proposes an improved methodology for code editing, which is a code intelligence task that involves modifying code based on natural language instructions.

**Theoretical Claims:**

- The reviewer has examined the proposed SeleKT algorithm for parameter optimization and considers it a general methodology for artificial neural network adaptation. However, the reviewer is uncertain whether its design choices are specifically tailored for the code editing task.

---

> ### Author Rebuttal · Authors · 2025-04-01
>
> We thank the reviewer for the insightful feedback on our work. We answer the questions asked by the reviewer below:
>
> > The scarcity of high-quality fine-tuning data and the risk of catastrophic forgetting during fine-tuning are well-known problems in ML models in general. This raises concerns that the paper’s contribution may not be sufficiently distinct.
>
> While we agree with the reviewer that catastrophic forgetting and lack of high quality finetuning data are well-known problems, we are unable to understand why this implies that our contribution is not sufficiently distinct.
>
> In fact, we do contrast and experimentally compare with some of the recent, key ML work in this space (PEFT/LoRA, model merging/TIES) throughout the paper. Our approach offers a conceptually novel solution. We introduce a novel weight update algorithm specifically designed to mitigate catastrophic forgetting during fine-tuning. Our strong results over model families, sizes, and benchmarks in the code-editing domain show improvements over standard techniques (SFT, LoRA, TIES).
>
> Further, our synthetic data generation pipeline also is a key contribution, that places emphasis on getting high-quality data tailored to the intricacies and diversity of real-world code-editing tasks and scenarios. Overall, our approach goes beyond existing methodologies and offers a nuanced, yet easy-to-implement approach to model adaptation.
>
> > The reviewer has examined the proposed SeleKT algorithm for parameter optimization and considers it a general methodology for artificial neural network adaptation. However, the reviewer is uncertain whether its design choices are specifically tailored for the code editing task.
>
> We agree with the reviewer about the potential generality of SeleKT. However, in this work, our motivation and focus is to improve the code-editing performance without sacrificing pre-learned abilities like code generation. The synthetic data generated for finetuning (Section 3) represents the design choices tailored specifically to the code-editing task. Code editing, and more generally coding models, is an important AI domain today, and our work clearly shows the method's potential. We note that in the present form, we have been careful not to make any claims about generality of SeleKT beyond what we demonstrate in the paper. We plan to investigate the applicability of SeleKT to more domains like math and natural language reasoning in the future.
>
> > The reviewer has a particular interest in the quality of the generated dataset. In Line 210 (L), the authors state that they perform quality-based filtering by prompting the LLM to select high-quality examples. However, the reviewer believes that this step requires some level of human intervention ... Conducting an empirical study to assess whether the LLM's evaluation standards align with those of human experts.
>
> Doing manual labeling for individual samples is impractical given the size of the synthesized dataset (100K's). We follow a long line of work in the literature on using LLM-as-a-judge to scale quality check.
>
> During the process of designing the synthetic data generation pipeline, we continuously monitored the quality of the generated data. Based on our observations, we implemented a stringent quality check process (Section 3, point iv; Appendix A.2, Figure 7) that filters out low-quality samples. Only instances meeting the specific criteria are retained. Recognizing the significant effort required for human expert labeling, our methodology provides a scalable alternative. Our experimental results clearly show improvements upon finetuning with the synthesized data. Nevertheless, following up on the reviewers' suggestion, we will conduct a study to evaluate agreement between the LLM and human reviewers on sample quality and include our findings in the paper.
>
>
> > A minor concern is whether the prompt length may exceed the model's context window (during data generation).
>
> Thank you for raising this concern. To clarify, we used GPT-4o and Llama-3.3-70B for data generation, both of which have sufficiently large context windows to accommodate our prompts. Therefore, exceeding the model’s context length was not an issue during the data generation process.

---

> > ### Comment · Reviewer_UeAD · 2025-04-04
> >
> > Despite the solid theoretical analysis of the proposed NextCoder method, given the high-stakes nature of the software engineering domain, the reviewer believes that additional evidence is needed to demonstrate the practical applicability of this conceptually novel solution. In particular, human-machine alignment analyses are encouraged for authors to be conducted. The reviewer would be willing to reconsider the score if a pilot human-alignment study or a post-generation case study is provided.

---

> > > ### Author Response · Authors · 2025-04-08
> > >
> > > Thank you for your valuable feedback. Based on the reviewer's suggestion, we conducted a pilot human-study to assess the quality of the generated training dataset. We are providing the detailed results below and would be happy to discuss them further.
> > >
> > > **Study Design**
> > >
> > > In this study, we involved three participants who have 3-4 years of experience in software development with strong expertise in Python. We randomly selected 100 samples from the Python split of our synthetic dataset and asked participants to rate each sample on the scale of 1-5 (1 being poor quality and 5 being excellent quality) on the following three questions:
> > >
> > >
> > > - **Q1 [Instruction Usefulness]**: How well does the detailed instruction capture a potential code-editing scenario with respect to the original code?
> > > - **Q2 [Instruction Consistency]**: How consistent are the three styles of instructions (detailed, concise and human-like) with each other and with the respective styles?
> > > - **Q3 [Solution Correctness]**: How well does the edited code match the edit described in the detailed instruction?
> > >
> > >
> > > **Overall Assessment**
> > >
> > > In the table below, we present the mean (along with standard deviations) ratings by participant and by question.
> > >
> > > | **Metric**               | **Participant 1** | **Participant 2** | **Participant 3** | **Overall Mean** | **Overall SD** |
> > > |--------------------------|-------------------------------|-------------------------------|-------------------------------|------------------|----------------|
> > > | Instruction Usefulness   | 4.92 ± 0.27                   | 4.93 ± 0.26                   | 4.38 ± 0.60                   | 4.74             | 0.48           |
> > > | Instruction Consistency  | 4.29 ± 0.48                   | 4.88 ± 0.32                   | 4.55 ± 0.54                   | 4.57             | 0.51           |
> > > | Solution Correctness     | 4.92 ± 0.27                   | 4.96 ± 0.24                   | 4.60 ± 0.51                   | 4.83             | 0.40           |
> > > | **Overall Mean ± SD**    | **4.71 ± 0.46**               | **4.92 ± 0.28**               | **4.51 ± 0.56**               |                  |                |
> > >
> > > The scores are consistently close to the highest score of 5 across all participants and questions. This provides a strong indication of human-machine alignment, with low to moderate variance. This study helps validate that our synthetic data generation pipeline is able to generate samples that meet human expectations in terms of quality and consistency. This complements the theoretical and empirical evidence we provide in the paper.
> > >
> > > We will incorporate this study into the revised version of the paper, along with selected qualitative examples, to further validate the design of our data generation pipeline and quality of the generated synthetic training data. We thank the reviewer for suggesting the AutoDSL [ACL'24] paper. The AutoDSL paper and "How to do human evaluation: A brief introduction to user studies in NLP" [NLE'23] cited therein were useful references towards conducting the human study.
> > >
> > > **Score Distribution**
> > >
> > > All samples received scores 3 (neutral quality) or above on all the questions. We give the exact distribution below.
> > >
> > > | **Score (Higher is better)** | **Instruction Usefulness** | **Instruction Consistency** | **Solution Correctness** |
> > > |----------:|----------------------------:|-----------------------------:|---------------------------:|
> > > | 5         | 229                         | 175                          | 250                        |
> > > | 4         | 65                          | 122                          | 48                         |
> > > | 3         | 6                           | 3                            | 2                          |
> > > | **Total** | 300                         | 300                          | 300                        |
> > >
> > >
> > > **Common Observations for Neutral Ratings (Score 3)**
> > >
> > > We particularly inspected the samples that received the neutral rating (score 3) since those were perceived as relatively low-quality samples by one or more participants. We made the following observations:
> > >
> > > - **Instruction-Edit Misalignment**: In some cases, instructions correctly described the intent but the edits were not entirely appropriate. For example, in response to an instruction to handle datetime parsing, the edited code parsed dates against raw strings, which would cause runtime errors.
> > > - **Incomplete Error Handling**: Some examples did introduce error handling, but overlooked edge cases (e.g., what if the `tasks.json` file exists but is empty?).
> > > - **Style Inconsistency**: A few participants noted that stylistic or structural variations across instruction formats led to minor misunderstandings of the code-editing intent.
> > >
> > > ---

---

### Official Review · Reviewer_3QHQ · 2025-03-13

**Overall Recommendation:** 4

**Summary:**

The paper introduces an adaptation method for code language models on code-edit tasks. The authors presents a synthetic data generation pipeline that creates code samples paired with edited versions and natural language instructions. The paper states that during fine-tuning, their SeleKT can update the model’s weights to avoid catastrophic forgetting and thus improve the performance. The authors also provides experiments to show that NextCoder outperforms comparable models on several code-editing benchmarks.

**Claims And Evidence:**

Claim: Selective adaptation via SeleKT improves editing performance without harming general code generation.
Evidence: Experiments on 4 benchmarks (Canitedit, Humanevalfix, Nofuneval, Aider), with comparisons against other methods, experiments shows that their model performs better consistently.

**Essential References Not Discussed:**

No.
However, it would benefit if the authors could add some discussion of recent studies on selective parameter updating and fine-tuning methods.

**Experimental Designs Or Analyses:**

The designs were validated via 4 benchmark datasets and other baseline methods, which seems fine.
However, it would be good if the authors could disscuss the reason for select some specific hyperparameters for seleKT.

**Methods And Evaluation Criteria:**

The approach to use a synthetic data pipeline to generate examples seems appropriate, the method that update the model weight seems working based on the experiments, the evaluations criteria also seems good.

**Other Comments Or Suggestions:**

Maybe the authors could add some discussion of potential trade-offs on hypermeter selection.

**Other Strengths And Weaknesses:**

Strength: The comparison table is clear and comprehensive, the 4 benchmark is sufficient to demonstrate the model's gain; overall, the paper is presented quite clear.

Weakness: limited theoretical analysis of the mechanism for ML interpretability, possible sensitivity in the hyperparameters

**Questions For Authors:**

What is the scalability and robustness of the approach on larger/smaller datasets? For instance, besides from codes and instructions, would there be possible application of this on proteins and explanations?

**Relation To Broader Scientific Literature:**

Their model, NextCoder, is based on previous work from Qwen2.5-Coder-7B, where they introduce new fine-tuning method to somewhat address catastrophic forgetting issue and resulted in a slightly better performance. Synthetic data generation and instruction tuning seems be related prior topics.

**Theoretical Claims:**

The paper does not seem to have formal theoretical proofs.

---

> ### Author Rebuttal · Authors · 2025-04-01
>
> We thank the reviewer for the insightful feedback on our work. We answer the specific questions below:
>
> > Reason for selecting some specific hyperparameters for seleKT.
>
> Our preliminary experiments indicated that a sparsity value of 5% and periodicity of 1 epoch were effective across model families, sizes, and benchmarks. Due to limited compute, we stick to these values in our experiments. Nevertheless, in Section 5.5, we present ablation on these two key hyperparameters of our method, namely, the sparsity $\alpha$ and periodicity $M$. By separately tuning the hyper-parameters for each model, we can get further improved performance for our approach. As suggested by the reviewer, we will include a discussion of hyperparameter selection and tradeoffs in the revision.
>
> > Robustness of the approach across dataset sizes
>
> In addition to answering the reviewer's question on (a) robustness to dataset sizes, we are also including new results on (b) robustness across model sizes.
>
> **(a) Robustness across dataset sizes**
>
> | Dataset size        | CanItEdit | HumanEvalFix | Aider |
> | --------------- | --------- | ------------ | ----- |
> | Base model (QwenCoder-2.5-7B) | 48.1 | 73.8 | 59.4 |
> | 25% | 47.67     | 80.20         | 60.90  |
> | 50%  | 48.57     | 81.43        | 62.70  |
> | 75% | 49.01     | 81.02        | 63.80  |
> | 100% (NextCoder-7B) | 50.48     | 81.10         | 65.70  |
>
> We appreciate the reviewer’s suggestion to evaluate effect of dataset size. To assess scalability w.r.t. to training data size, we finetuned the QwenCoder-2.5-7B model on varying fractions (random sampling 25%, 50% and 75%) of our dataset which includes both synthetic and CommitPackFT data. The results are presented in the table above. All models were trained for 3 epochs. The results show a clear trend: while performance on CanItEdit and Aider sees a drop at 25% w.r.t to the base model, increasing the dataset size consistently improves performance across all benchmarks (CanItEdit, HumanEvalFix, and Aider).
>
>
> **(b) Robustness across model sizes**
>
> Additionally, we are happy to share new results (the tables below) comparing the performance of our SeleKT algorithm across various model sizes against supervised finetuning (SFT) and parameter-efficient low-rank adaptation (LoRA) on multiple code editing benchmarks.
>
> | **3B Models** | **HumanEvalFix** | **CanItEdit** | **Aider** |
> |:----------|:----------------:|:-------------:|:---------:|
> | QwenCoder-2.5-3B | 73.2 | 37.1 | 36.8 | - |
> | QwenCoder-2.5-3B-LoRA | 64.6 | 36.2 | 35.8 | - |
> | QwenCoder-2.5-3B-SFT | **76.2** | 32.4 | 30.1 | - |
> | **NextCoder-3B** | 75.6 | **42.4**|**37.6** | - |
>
> | **14B Models** | **HumanEvalFix** | **CanItEdit** | **Aider** | **Polyglot** |
> |:----------|:----------------:|:-------------:|:---------:|:------------:|
> | QwenCoder-2.5-14B | 87.8 | 58.1 | 66.9 | 9.3 |
> | QwenCoder-2.5-14B-LoRA | 78.0 | 50.9 | 66.2 | 5.3 |
> | QwenCoder-2.5-14B-SFT | 79.9 | 42.4 | 36.8 | 3.1 |
> | **NextCoder-14B** | **89.8** | **60.2** | **72.2** | **12.2** |
>
> | **32B Models** | **HumanEvalFix** | **CanItEdit** | **Aider** | **Polyglot** |
> |:----------|:----------------:|:-------------:|:---------:|:------------:|
> | QwenCoder-2.5-32B | **90.2** | 61.0 | 72.9 | 16.4 |
> | QwenCoder-2.5-32B-LoRA | 82.3 | 52.4 | 60.2 | 6.7 |
> | QwenCoder-2.5-32B-SFT | 81.7 | 49.5 | 66.9 | 8.4 |
> | **NextCoder-32B** | 88.9 | **62.4** | **74.7** | **21.9** |
>
> For the smaller 3B model, NextCoder-3B shows significant improvements over the base model across most benchmarks, with a substantial gain on the CanItEdit benchmark (+5.3\%).
>
> For larger models, we have also included the latest and more challenging[ Aider-Polyglot benchmark](https://aider.chat/docs/leaderboards/) results (more details in response to reviewer itGF).
>
>
> > Limited theoretical analysis of the mechanism
>
> Though the focus of our paper is on rigorous empirical validation, we have made some progress on theoretical understanding, that we outline next. Under the standard smoothness assumption on $f$ and boundedness assumptions on the gradients and concentration assumption on the task vector stated below, we can show $O(1 /\sqrt{T})$ convergence for SeleKT. Due to char limits, we are unable to give a proof sketch here.
>
> **Task Vector Concentration Assumption**
>
> The task vector $\tau = \theta - \theta_{base}$ exhibits a concentration property with parameter $r >> 1$, such that the majority of information is contained in the top-$\alpha$ fraction of parameters.
>
> $$
> \frac{{\sum_{i \in \text{top-}\alpha} |\tau_i|^2}}{\alpha N} \geq r \cdot \frac{{\sum_{i \notin \text{top-}\alpha} |\tau_i|^2}}{(1-\alpha)N} , \quad r >> 1
> $$
>
> Note that $r$ defines a certain margin between the task-specific parameters (i.e. top-$\alpha$ fraction of parameters) and the rest of the parameters (both suitably normalized).
>
> > Besides from codes and instructions, would there be possible application...
>
> Please refer to our response to reviewer UeAD (2nd point).

---

### Decision · Program_Chairs · 2025-05-01

**Decision:**

Accept (poster)

**Comment:**

This work presents a synthetic data generation pipeline for code editing along with a method (seleKT) for “robust” LLM adaptation. The evaluation, along with some results presented during the rebuttal, suggest that the data/adaptation method performs strongly against alternatives.

The method presented is reasonable, the problem important, and the evaluation results are positive and hence I recommend that this paper is accepted. Please, incorporate the relevant results discussed in the rebuttal.

Beyond that, I should note that it’s not clear to me why the contributions of SeleKT and code editing data appear in this paper as they seem somewhat independent. In particular, SeleKT could have been evaluated in a wider range of applications and adaptation scenarios. I suggest that the authors either state more explicitly why SeleKT is only well-suited for code editing adaptation or provide some additional validation about the generalizability (or not) of SeleKT.